# 🌐BESPOKE: BENCHMARK FOR SEARCH-AUGMENTED LARGE LANGUAGE MODEL PERSONALIZATION VIA DIAGNOSTIC FEEDBACK

## ABSTRACT

Search-augmented large language models (LLMs) have advanced information-seeking tasks by integrating retrieval into generation, reducing users' cognitive burden compared to traditional search systems. Yet they remain insufficient for fully addressing diverse user needs, which requires recognizing how the same query can reflect different intents across users and delivering information in preferred forms. While recent systems such as ChatGPT and Gemini attempt personalization by leveraging user histories, systematic evaluation of such personalization is under-explored. To address this gap, we propose BESPOKE, the realistic benchmark for evaluating personalization in search-augmented LLMs. BESPOKE is designed to be both realistic, by collecting authentic chat and search histories directly from humans, and diagnostic, by pairing responses with fine-grained preference scores and feedback. The benchmark is constructed through long-term, deeply engaged human annotation, where human annotators contributed their own histories, authored queries with detailed information needs, and evaluated responses with scores and diagnostic feedback. Leveraging BESPOKE, we conduct systematic analyses that reveal key requirements for effective personalization in information-seeking tasks, providing a foundation for fine-grained evaluation of personalized search-augmented LLMs. Our code and data are available at `https://anonymous.4open.science/r/bespoke-E82B/`.

## 1 INTRODUCTION

Information-seeking tasks aim to address a user's information need by providing the desired information in an appropriate form. Recently, search-augmented LLMs have driven a fundamental shift in information-seeking tasks, leveraging the retrieval-augmented generation (RAG). Unlike traditional search systems that simply list relevant content and require users to read, compare, and synthesize information themselves (Ji et al., 2024; Juneja et al., 2024), RAG reduces cognitive burden by integrating information into responses, thereby more effectively addressing users' information needs.

Nevertheless, these advances remain insufficient for fully addressing users' information needs. To genuinely achieve this, a system should accurately identify how the query can reflect different needs across users by considering their backgrounds (Salemi & Zamani, 2025a), and ensure that the information is delivered in a form that matches their preferences (Kumar et al., 2024; Salemi et al., 2025). As shown in figure 1, although both users issue the same query, their different backgrounds lead one to focus on *"environmental implication"* while the other emphasizes *"performance"*. Moreover, they prefer different forms of information delivery, with one favoring a narrative explanation and the other a concise bullet summary. Since RAG cannot fully satisfy these requirements alone, there is a growing need for systems that can adapt to diverse user intents and presentation preferences.

In response to these needs, recent search-augmented LLMs such as ChatGPT (OpenAI, 2023) and Gemini (Gemini-Team, 2025) have moved beyond generic outputs by leveraging users' prior chat and search histories as user contexts to personalize their responses. This shift enables them to better reflect each user's distinct preferences, such as topic interests, informational needs, and delivery style. However, despite this advancement, systematic evaluation of these systems to diagnose limitations and guide further development remains largely underexplored.

Figure 1: Personalization in a search-augmented LLM. It infers user-specific needs from history, conducts tailored search, and adapts the delivery of information to align with user preferences.

In this paper, we introduce BESPOKE, a realistic benchmark specifically designed for evaluating personalization in search-augmented LLMs. BESPOKE is built from 2,870 real user history sessions collected over three weeks, where human annotators with diverse backgrounds engaged in natural conversations and web searches as part of their daily routines. Grounded in these fully human-annotated histories, BESPOKE additionally provides 150 user-annotated queries and their corresponding gold information needs, which explicitly outline the personalized requirements for each query. Unlike existing personalization benchmarks for information-seeking tasks, such as LaMP-QA (Salemi & Zamani, 2025a), which remain limited to QA-style interactions in constrained domains, BESPOKE encompasses a broad spectrum of user activities on the web, including free-form chats that extend beyond information seeking and actual web-search histories.

In addition to providing gold information for each query, BESPOKE offers response-judgement pairs with human-annotated scores and explanatory feedback, explicitly clarifying why a response is deemed satisfactory or unsatisfactory. Based on these, we propose an evaluation framework that effectively assesses personalization in information-seeking tasks. By leveraging human-annotated feedback as additional context, our evaluation framework achieves better alignment with human judgment. Moreover, it not only assesses whether responses factually and accurately incorporate relevant information but also evaluates whether personalization is well achieved, delivering both scores and diagnostic feedback. This provides a detailed diagnosis, which serves as supervision for developing personalized systems (Balepur et al., 2025; Salemi & Zamani, 2025b).

With BESPOKE, we provide an extensive analysis of personalization in search-augmented LLMs, offering insights into how effective personalization can be achieved in information-seeking. Our results show that personalization is strongly influenced by how user contexts are constructed from user histories, while many models still fall short in delivering personalized responses. These findings highlight BESPOKE as a challenging benchmark and emphasize the need for more effective methods.

The main contributions of our work are summarized as follows:

- We propose BESPOKE, the benchmark that genuinely reflects real-world personalization scenarios in search-augmented LLMs through long-term, deeply engaged human annotation.
- BESPOKE provides an evaluation framework that offers diagnostic feedback under fine-grained criteria, enabling specific evaluation of personalization and guiding system improvements.
- Leveraging BESPOKE, we provide systematic analyses that highlight essential requirements for achieving effective personalization in information-seeking tasks.

## 2 RELATED WORK

**Search-augmented LLM evaluation.** Search-augmented LLMs, which leverage RAG, have recently driven a fundamental shift in information-seeking, enabling more factual and robust outputs (Gao et al., 2023b; Huang & Huang, 2024). To evaluate such systems, several methods (Ru et al., 2024; Park et al., 2025a) have been proposed, providing controlled settings that diagnose generation errors and test robustness to noisy contexts. However, these frameworks primarily rely on simple fact-based QA queries and consequently assess RAG systems mainly from the perspective of factuality. Another recent effort, RAG-QA Arena (Han et al., 2024), evaluates long-form

Table 1: Comparison with existing information-seeking benchmarks.

| Dataset | Personalization | Unconstrained Domain | History Type | Preference Annotation | Gold Info. Annotation |
|---|---|---|---|---|---|
| RAG-QA Arena (Han et al., 2024) | ✗ | ✗ | ✗ | Chosen/Reject | ✓ |
| Search-Arena (Miroyan et al., 2025) | ✗ | ✓ | ✗ | Chosen/Reject | ✗ |
| LaMP-QA (Salemi & Zamani, 2025a) | ✓ | ✗ | QA-pair | ✗ | ✗ |
| **BESPOKE (ours)** | ✓ | ✓ | **Chat + Search history** | **Score+Feedback** | ✓ |

QA by pairwise preference judgments, but the evaluation scope remains limited to specialized domains. Search Arena (Miroyan et al., 2025) broadens the scope to open-web sources and diverse query types, reflecting the growing adoption of open-setting evaluation as the standard paradigm for assessing information-seeking tasks (Gou et al., 2025; Kasai et al., 2024). Similar to RAG-QA Arena, it introduces a pairwise evaluation framework that collects head-to-head human preference judgments between system outputs, allowing for comparative analysis of search-augmented LLMs under realistic search tasks. While Search Arena provides a strong foundation, it remains limited to general preference judgments without incorporating personalized aspects of evaluation.

**Personalized LLM evaluation.** Recent work on personalized LLM evaluation has investigated whether models can adapt outputs to user attributes or preferences. Early benchmarks (Salemi et al., 2023; Kumar et al., 2024) use synthetic personas, and LaMP-QA (Salemi & Zamani, 2025a) extends this line to information-seeking tasks by constructing a dataset from StackExchange, pairing queries with user profiles from past questions. However, relying on platforms such as StackExchange constrains the dataset domain and restricts user histories to QA-style interactions. Taken together, these studies highlight two essential components for personalized evaluation: leveraging realistic user histories to represent preferences and reasoning over histories to infer information needs.

## 3 CONSTRUCTING BESPOKE

We introduce BESPOKE, a realistic and diagnostic benchmark that evaluates personalization in search-augmented LLMs with diagnostic feedback. To collect sufficient user histories and detailed feedback, we employ a long-term, deeply engaged human annotation. Over 3 weeks, annotators freely engage in diverse activities like information-seeking and chatting, accumulating their own chat and web-search histories. Then they issue queries grounded in the information needs arising from these histories and provide preference scores and feedback on the sampled responses generated for the queries by search-augmented LLMs. The overall pipeline is illustrated in Figure 2.

Table 2: Statistics of BESPOKE. The diversity index uses Shannon's equitability (Appendix B.1), with values near 1 indicating higher user diversity.

| Statistics | Value |
|---|---|
| #Users (Div. index) | 30 (0.91) |
| #Sessions | 2,870 |
| – Search session | 2,153 |
| – Chat session | 717 |
| #Avg. sesssion / user | 95.67 |

### 3.1 TASK FORMULATION

For a user $u$ issuing a query $q$, we define user history as $\mathcal{H}_u = \{\mathcal{S}_u, \mathcal{C}_u\}$, consisting of their search history $\mathcal{S}_u$ and chat history $\mathcal{C}_u$. A search-augmented LLM begins by inferring the user's information need $n_q$ implied by $q$ either explicitly or implicitly through their history $\mathcal{H}_u$. Guided by the inferred $n_q$, the model then searches for relevant information. By using both searched information and $\mathcal{H}_u$, it subsequently generates a response $r$ to address $n_q$ in a manner that aligns closely with the user's judgments. Under this task formulation, BESPOKE evaluates how accurately $r$ conveys the information the user seeks through $q$ and how well it is personalized to the user.

### 3.2 HUMAN ANNOTATOR & HISTORY COLLECTION

**Human annotator recruitment.** The first step of constructing BESPOKE is to recruit human annotators. Since personalized responses depend on the user's unique characteristics, it is essential that the benchmark reflects a broad range of real-world user characteristics to meaningfully evaluate a model's ability to adapt across diverse user contexts. To achieve this, we recruit 30 human annotators with diverse backgrounds across professions and interests, ensuring these backgrounds cover a wide range of interaction patterns and contexts. More details are provided in Appendix B.1.

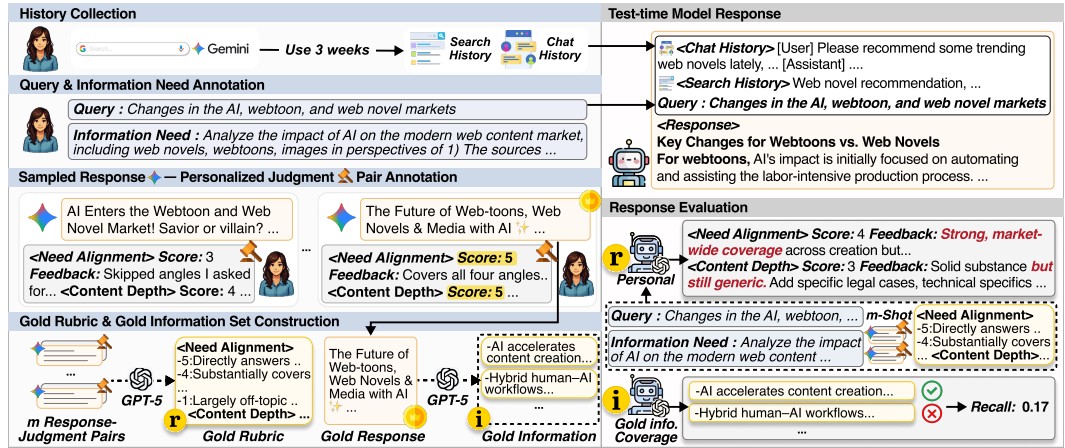

Figure 2: Overview of our BESPOKE construction process.

**History collection.** Recent systems such as Gemini (Gemini-Team, 2025) have demonstrated the value of leveraging both chat and search histories to personalize search-augmented LLM. Motivated by this development, BESPOKE incorporates both search histories $\mathcal{S}_u$ and chat histories $\mathcal{C}_u$ as complementary sources of user context. To obtain these user histories, each annotator is assigned a dedicated Google account created solely for this study. During the history collection period, they freely use this account in their daily lives for information seeking, conducting Google searches, and conversing with Gemini (Gemini-Team, 2025) according to their own interests and routines. We set the collection period to three weeks to ensure that enough interaction data is accumulated to reflect implicit user preferences. At the end of the history collection period, we collect the resulting histories after carefully removing all personally identifiable or sensitive information. For details on data handling and privacy protection, please refer to the Appendix B.2.

## 3.3 HUMAN ANNOTATION PIPELINE

After collecting users' search and chat histories, we employ a multi-stage annotation process aimed at constructing user queries and obtaining fine-grained feedback on the model responses, encompassing both preference scores and feedback. This process consists of three stages: 1) authoring a simple query together with its gold information need, 2) annotating multiple responses with both numerical scores and verbalized feedback, and 3) manually generating a gold response. For more details about the human annotation pipeline, please refer to the Appendix B.3.

**Task types.** Among the intent categories proposed in Search-Arena (Miroyan et al., 2025), we select four tasks: *analysis*, *guidance*, *recommendation*, and *explanation*, as the task types in BESPOKE. These tasks represent information-seeking scenarios where personalization is essential since the same information may need to be delivered in a different tone or style depending on the user.

**Query & Gold information need.** Each annotator is asked to issue five test queries, covering all four task types. For each of the five test queries, annotators are asked to provide two annotations: a simple query $q$ and a corresponding gold information need $n_q^+$. The simple query reflects how the user would ask in daily life, assuming that the system will infer the underlying information need from prior interaction history. In contrast, the gold information need is written in a more detailed form than the simple query, explicitly stating their background, intent, and the specific aspects to be addressed, ensuring that the desired information is clearly specified.

**Response-Judgment pair (R-J pair).** For each query $q$, we sample $k$ sets of response $r$, which the same annotator then assigns a five-point Likert score $s$ with diagnostic feedback $f$ over 4 criteria:

- **Need alignment**: Assesses how well the response captures the user's information need, focusing on whether it directly addresses the desired content while avoiding irrelevant information.
- **Content depth**: Evaluates the level of detail and complexity in the response. It assesses whether the explanation is thorough, concise, or advanced, given the user's expectations and the query.
- **Tone**: Measures how well the response's tone (e.g., formal, casual) matches the user's judgments.

- **Explanation style**: Examines whether the structure, flow, and illustrative methods, such as step-by-step guidance or examples, align with the user's preferred way of understanding information.

The judgment $(s, f)$ captures user judgments and provides a detailed assessment of personalization quality, highlighting which aspects of a response align with or deviate from the user's expectations. For each query $q$, this results in a set of $k$ response–judgment tuples $(r, s, f)$. To obtain such diverse assessments, we adopt a two-stage process for collecting judgments from each annotator.

In the first stage, the goal is to capture a spectrum of user judgments by having annotators evaluate multiple candidate responses for each query $q$. Specifically, for each query $q$, we construct a set of $k$ information needs, consisting of the gold need $n_q^+$ and $k - 1$ additional hypothetical needs plausibly inferred from $q$ using Gemini-2.5-Pro (Gemini-Team, 2025). For each of these $k$ information needs, Gemini-2.5-Pro-Grounding is used to produce a response grounded on Google Search. Annotators then evaluate all $k$ responses along the four criteria, resulting in $k$ R-J pairs.

In the second stage, the goal is to construct a high-quality gold response $r^+$. Annotators first search for passages relevant to $n_q^+$ using Google Search, then prompt Gemini-2.5-Pro grounded on those passages to draft a response. They iteratively provide scores and feedback on the four criteria, and Gemini refines the draft accordingly. This process continues until the output fully satisfies all four criteria with the maximum score of 5, which defines the gold response $r^+$. During this refinement process, additional R-J pairs are also obtained as byproducts of the iterative updates.

## 3.4 EVALUATION

Building upon the human-annotated data described in Section 3.3, our evaluation framework employs LLMs to evaluate generated responses following Zheng et al. (2023); Liu et al. (2023). In particular, we adopt GPT-5 as the backbone to mitigate self-preference bias using a model distinct from Gemini, which is used in the human annotation process (Panickssery et al., 2024).

Our primary evaluation focuses on *personalization quality*, assessed along the four criteria defined in Section 3.3: need alignment, content depth, tone, and explanation style. To be specific, a GPT-5-based evaluator $\mathcal{E}_p$ operates in a few-shot setting to evaluate a new response $\hat{r}$ for a given query $q$. For each query $q$, we first construct query-specific demonstration examples $\mathcal{D}_q = \{(r_i, s_i, f_i)\}_{i=1}^m$ based on the R-J pairs collected in Section 3.3, where each tuple consists of a response $r_i$, its scalar score $s_i$, and diagnostic feedback $f_i$ for a specific criterion. From these examples in $\mathcal{D}_q$, we then leverage GPT-5 to generate a query-specific gold rubric $\mathcal{R}_q^+$, which provides personalized evaluation guidelines scaled from 1 to 5 points per criterion. Finally, the evaluator $\mathcal{E}_p$ incorporates $\mathcal{D}_q$, $\mathcal{R}_q^+$, and the gold information need $n_q^+$ to produce a scalar score $s$ and diagnostic feedback $f$ for each criterion on the new response $\hat{r}$: $(s, f) = \mathcal{E}(\mathcal{D}_q, \mathcal{R}_q^+, n_q^+, q, \hat{r})$.

To complement personalization criteria, we also measure *gold information coverage*, which evaluates how well a response reflects the *gold information* $\mathcal{I}_q^+$ at the level of atomic claims. Specifically, from each $r^+$, we first use GPT-5 to extract atomic claims as a candidate gold information set. We then manually inspect these candidates and retain only those verifiable via web search, which collectively constitute the gold information set $\mathcal{I}_q^+ = \{i_{q,1}^+, \ldots, i_{q,n}^+\}$. Then, given a model response $\hat{r}$, a GPT-5-based evaluator $\mathcal{E}_r$ checks each $i_{q,j}^+ \in \mathcal{I}_q^+$ via binary classification to determine whether it is correctly expressed in $\hat{r}$ without contradiction. Let $\mathcal{I}_{\hat{r}}$ denote the set of gold claims judged as present in $\hat{r}$. Recall$(\hat{r})$ is then computed as $|\mathcal{I}_{\hat{r}}|/|\mathcal{I}_q^+|$, which directly measures whether gold information is conveyed in the response. This formulation assesses whether the gold information is accurately presented in the response, making it suitable for open-web settings where information may appear redundant or include irrelevant content. For more details, please refer to the Appendix C.1.

## 4 EXPERIMENTS

**Experimental settings.** We evaluate 6 representative search-augmented LLMs: GPT-4o-search, o3-search, Gemini-2.5-Flash, Gemini-2.5-Pro, Perplexity-sonar and Perplexity-sonar-reasoning. For personalization, the $\mathcal{E}_p$ assesses responses across four criteria mentioned in Section 3.3. Personalization scores are reported on a 0–100 scale by multiplying raw 1–5 Likert ratings by 20, following

Zhu et al. (2025). For gold information coverage, recall is assessed by $\mathcal{E}_r$ as the proportion of gold information correctly expressed in a response. For more details, please refer to Appendix C.1.

## 4.1 META EVALUATION

Before the main experiments, we validate our GPT-5-based evaluator[1] $\mathcal{E}_p$ through a meta-evaluation, confirming that its judgments align with those of human annotators.

**Dataset and Baselines.** We construct a meta-eval dataset composed of 300 R–J pairs that are not used in the evaluator's demonstration shots, serving as a reference against human judgments. Specifically, for each of the 30 users, we sample two R–J pairs per query. We compare our evaluator $\mathcal{E}_p$ (w/ Feedback) against commonly used LLM-based evaluator baselines: a generic one without personalization (w/o Personal.) and one that provides scores only without feedback as context (w/o Feedback). For more details, please refer to the Appendix C.2.

**Evaluation process.** With the meta-evaluation dataset, we validate whether our $\mathcal{E}_p$ reliably reproduces human judgments for our four personalization criteria on the generated responses. Concretely, we assess the evaluator along two dimensions: whether it assigns scores consis-

Table 3: Meta evaluation result. Pearson / Spearman correlation shows the agreement with human preference judgments, and Feedback Acc. represents the alignment with human feedback.

| $\mathcal{E}_p$ Type | Criterion | Pearson Corr. | Spearman Corr. | Feedback Acc. |
|---|---|---|---|---|
| w/o Personal. | Need Align. | 0.374 | 0.373 | 0.160 |
| | Content depth | 0.356 | 0.385 | 0.270 |
| | Tone | 0.639 | 0.625 | 0.677 |
| | Style | 0.474 | 0.485 | 0.300 |
| | Avg. | 0.461 | 0.467 | 0.352 |
| w/o Feedback | Need Align. | 0.844 | 0.849 | 0.803 |
| | Content depth | 0.814 | 0.825 | 0.767 |
| | Tone | 0.874 | 0.852 | 0.827 |
| | Style | 0.689 | 0.712 | 0.727 |
| | Avg. | 0.805 | 0.810 | 0.781 |
| w/ Feedback (ours) | Need Align. | 0.881 | 0.882 | 0.850 |
| | Content depth | 0.845 | 0.855 | 0.870 |
| | Tone | 0.900 | 0.893 | 0.903 |
| | Style | 0.784 | 0.797 | 0.857 |
| | Avg. | **0.853** | **0.857** | **0.870** |

tent with human judgments (*score alignment*) and whether it produces feedback semantically similar to human feedback (*feedback alignment*). For score alignment, we compute the Pearson correlation and Spearman correlation between the evaluator's scores and the human annotators' scores on the same responses. Feedback alignment is then assessed by comparing annotator-written feedback with evaluator-generated feedback for the same response using an LLM-as-a-judge meta-evaluator. Specifically, using GPT-5 as a meta-evaluator, we determine if the two feedbacks convey the same content given the context $(q, n_q^+, \mathcal{R}_q^+)$. The meta-evaluator then outputs a binary label indicating feedback equivalence, and accuracy is defined as the proportion of cases judged equivalent.

**Evaluation results.** As presented in Table 3, evaluator baselines without personalization yield low correlations with human judgments. In contrast, our evaluator $\mathcal{E}_p$ demonstrates strong agreement with human judgments, clearly confirming its ability to assess responses in a manner highly similar to humans. This suggests that personalized evaluation requires more than a generic evaluator and instead calls for a framework designed specifically for personalization. Notably, incorporating diagnostic feedback into the evaluator's context yields superior alignment with human judgments compared to baselines that rely solely on scores, underscoring the critical role of human-annotated feedback in enhancing personalization evaluators. Similarly, for feedback alignment, our approach shows strong alignment with human feedback, consistently outperforming the baselines across all criteria. These results suggest that the human-annotated feedback provided by BESPOKE makes a significant contribution to building an effective personalization evaluator.

## 4.2 EFFECT OF USER CONTEXT

We investigate whether leveraging user history helps personalization in search-augmented LLMs and how user context can be most effectively constructed from it. To this end, we vary four aspects of user context construction. 1) *History usage* determines whether the search-augmented LLM is provided with user context constructed from user history or without it. 2) *Query awareness* specifies whether the user context is kept fixed across all queries from a given user or dynamically constructed for each query. In query awareness setting, 3) *history selection* determines whether to use the entire history or only query-relevant histories, with relevance assessed by an LLM through

---

[1] Although our main evaluator is built with GPT-5, we observe similarly high agreement with human judgments when replacing it with strong open-weight models. Please refer to Appendix A.1

Table 4: Evaluation of search-augmented LLMs under different configurations. The best and second-best results are shown in **bold** and underline. Experimental details are provided in Appendix C.3.

| Model | History Usage | Query Awareness | History Selection | User Context | Need Align. | Content Depth | Tone | Style | Recall | Avg. |
|---|---|---|---|---|---|---|---|---|---|---|
| ⊛ pplx-sonar | ✗ | ✗ | ✗ | ✗ | 45.20 | 47.47 | 80.53 | 56.53 | 9.87 | 47.92 |
| | ✓ | ✗ | ✗ | Profile | 46.00 | 48.00 | 80.13 | 62.80 | 11.89 | 49.76 |
| | ✓ | ✓ | ✗ | Profile | 52.53 | 56.93 | 84.67 | 71.47 | 23.25 | 57.77 |
| | ✓ | ✓ | ✓ | Raw | 47.33 | 50.20 | 79.73 | 60.53 | 11.23 | 49.79 |
| | ✓ | ✓ | ✓ | Profile | **55.80** | **59.90** | **85.13** | **72.37** | **25.50** | **59.74** |
| ⊛ pplx-sonar -reasoning | ✗ | ✗ | ✗ | ✗ | 46.53 | 50.27 | 75.87 | 62.53 | 11.11 | 49.26 |
| | ✓ | ✗ | ✗ | Profile | 47.87 | 51.07 | 81.20 | 65.47 | 14.34 | 51.99 |
| | ✓ | ✓ | ✗ | Profile | 49.87 | 54.80 | 80.00 | 70.00 | 21.92 | 55.32 |
| | ✓ | ✓ | ✓ | Raw | 46.40 | 50.67 | 76.00 | 61.60 | 11.58 | 49.25 |
| | ✓ | ✓ | ✓ | Profile | **54.27** | **57.47** | **83.33** | **70.67** | **23.93** | **57.93** |
| ✦ Gemini-2.5 flash-grounding | ✗ | ✗ | ✗ | ✗ | 47.87 | 50.80 | 79.47 | 62.27 | 15.42 | 51.16 |
| | ✓ | ✗ | ✗ | Profile | 49.73 | 52.60 | 78.30 | 66.33 | 17.25 | 52.84 |
| | ✓ | ✓ | ✗ | Profile | 52.93 | 57.60 | 82.67 | 70.27 | 27.14 | 58.12 |
| | ✓ | ✓ | ✓ | Raw | 49.63 | 52.13 | 77.20 | 62.13 | 16.08 | 51.43 |
| | ✓ | ✓ | ✓ | Profile | **55.73** | **61.03** | **82.83** | **71.73** | **28.09** | **59.88** |
| ✦ Gemini-2.5 pro-grounding | ✗ | ✗ | ✗ | ✗ | 47.60 | 50.40 | 74.67 | 62.13 | 12.83 | 49.53 |
| | ✓ | ✗ | ✗ | Profile | 49.07 | 54.67 | 80.40 | 68.40 | 16.67 | 53.84 |
| | ✓ | ✓ | ✗ | Profile | 54.83 | **60.33** | 83.33 | 73.00 | **25.41** | 59.38 |
| | ✓ | ✓ | ✓ | Raw | 51.60 | 51.87 | 79.73 | 65.60 | 15.00 | 52.76 |
| | ✓ | ✓ | ✓ | Profile | **56.40** | 60.27 | **84.40** | **72.40** | 25.32 | **59.76** |
| ⑨ gpt-4o-search | ✗ | ✗ | ✗ | ✗ | 44.23 | 46.43 | 79.67 | 59.33 | 8.13 | 47.56 |
| | ✓ | ✗ | ✗ | Profile | 44.17 | 46.77 | 81.33 | 62.37 | 9.23 | 48.77 |
| | ✓ | ✓ | ✗ | Profile | 50.27 | 53.07 | 84.80 | 68.93 | 16.24 | 54.66 |
| | ✓ | ✓ | ✓ | Raw | 45.97 | 48.07 | 81.00 | 61.37 | 9.17 | 49.11 |
| | ✓ | ✓ | ✓ | Profile | **53.80** | **57.20** | **84.83** | **69.93** | **19.23** | **57.00** |
| ⑨ o3-search | ✗ | ✗ | ✗ | ✗ | 51.60 | 57.47 | 78.53 | 70.00 | 22.05 | 55.93 |
| | ✓ | ✗ | ✗ | Profile | 54.53 | 60.53 | 75.87 | 71.60 | 25.73 | 57.65 |
| | ✓ | ✓ | ✗ | Profile | 55.73 | 61.87 | 84.80 | 74.80 | 28.61 | 61.16 |
| | ✓ | ✓ | ✓ | Raw | 55.40 | 60.60 | 78.27 | 71.40 | 23.88 | 57.91 |
| | ✓ | ✓ | ✓ | Profile | **59.07** | **63.73** | **85.20** | **73.87** | **30.53** | **62.48** |

pairwise evaluation of the query against each history session. (Weller et al., 2025). We adapt this approach for our experiments, given the inherent limitations of bi-encoders discussed in Section 4.4. 4) *User context form* refers to how user contexts are presented to the model, either as raw histories or as profiles structuring user preferences. In our experiment, we consider two types of profiles: *general profile* (*Query Awareness* ✗) and *query-aware profile* (*Query Awareness* ✓). General profile summarizes a user's entire search and chat history to capture stable, long-term preferences such as topical interests, preferred information formats, tone, and explanation styles. While effective for modeling broad tendencies, such profiles can dilute fine-grained or context-specific preferences. query-aware profiles address this limitation by selecting only the subset of history sessions that are relevant to the current query before summarization. More details are provided in Appendix C.3.

Table 4 shows our experimental results. Overall, leveraging user history consistently improves personalization performance across all models, though the extent of improvement varies by criterion.

**Inferring user needs and delivering appropriate information remain challenging.** For need alignment, content depth, and recall, the baseline performance is considerably lower. This highlights the difficulty of inferring a user's information need from a query and searching information that adequately addresses it. Notably, recall remains substantially lower than the other two dimensions, suggesting that providing users with precise and detailed information remains highly challenging. However, these three dimensions show clear improvements once user contexts are incorporated, suggesting that leveraging user history enables search-augmented LLMs to better infer users' information need and deliver detailed information at a level appropriate to their knowledge.

**Personalization enables finer alignment of tone and style with user preferences.** In particular, models already achieve relatively high scores for tone and style even without personalization. This is supported by our analysis based on the feedback presented in Section 4.3, indicating that users generally prefer the neutral and concise tone and style of default LLM outputs in information-seeking. Nevertheless, personalization for tone and style remains necessary, as leveraging user history yields substantial gains and enables models to more precisely match user preferences.

**Query : What is a balanced diet?**
**Information need :** *If I want to follow a balanced diet, what factors should I consider? Is it possible to maintain a balanced diet while dieting? And if I pay attention to my diet, can I get all the nutrients my body needs without taking supplements like vitamins?*

| GPT-4o Response w/ Full History Profile | Bespoke Score & Feedback |
|---|---|
| A balanced diet involves consuming the right proportions of macronutrients—carbohydrates, proteins, and fats—as well as ensuring sufficient intake of essential vitamins and minerals. **### Macro Ratios and Nutrient Balance:** - **Carbohydrates (4:4:2 or 5:3:2 ratios):** Aim for complex carbs like brown rice or sweet potatoes, ensuring a steady.. | *<Need Alignment> 2*, On-topic **but too narrow. It focuses on macro percentages only.** I needed food-first specifics (protein targets, carb ... *<Content Depth> 2*, **Too basic—mostly ratios without usable numbers** or examples. Add portions/kcal, simple targets (e.g., protein g/kg, ≥... *<Tone> 4*, Neutral and concise; easy to read. It would feel more practical with ... *<Explanation Style> 2*, **Missing the principles→examples→implementation flow.** Please add a plate-builder template and a sample day with ... |
| **GPT-4o Response w/ Selected History Profile** | **Bespoke Score & Feedback** |
| A balanced diet provides your body with the essential nutrients it needs to function optimally while supporting your weight loss goals through a sustainable calorie deficit. **Core Principles:** - **Protein at Each Meal:**Incorporate lean protein sources to promote satiety and preserve muscle mass. ... - **Carbohydrates: 45–65%** of total daily calories - **Protein: 10–35%** of total daily calories ... | *<Need Alignment> 4*, **Good, on-topic, and actionable**—nice plate method, build-a-meal template, sample day, and protein target for dieting.... *<Content Depth> 4*, **Clear and practical with macro ranges**, 1.6–2.2 g/kg protein, and a 25–35 g fiber target. It needs portions and kcal ... *<Tone> 4*, Neutral and crisp. The closing pep-talk ("You've got this!") isn't necessary—slightly tighter, purely informational ... *<Explanation Style> 4*, **Solid principles → templates → sample day flow** and an actionable checklist. Add a concise supplements section... |

Figure 3: Comparison of GPT-4o-search responses in the lowest/highest performance setting, illustrating how feedback captures difference in personalization quality. More examples in Appendix F.

**Query-aware user contexts improve personalization.** Performance also varies substantially depending on how user histories are leveraged to construct the user context. In particular, it is more effective to build contexts in a query-aware manner (✓ *in query awareness*) rather than relying on static user context (✗ *in query awareness*). This improvement occurs since a single user may prioritize different personalization directions depending on the query topic, such as emphasizing professional background in work-related queries but personal interests in leisure-related ones.

**Selective history selection boosts context relevance.** Furthermore, within query-aware approaches, selectively choosing relevant portions of user history (✓ *in history selection*) proves more effective than using the entire history indiscriminately (✗ *in history selection*). Indiscriminate inclusion can introduce irrelevant or outdated information that weakens the focus, whereas selective inclusion filters for the most pertinent interactions, ensuring the user context is concise and directly applicable to the current query. This targeted approach allows models to better extract and utilize key signals from the history, leading to more precise personalization.

**Structured profiles outperform raw histories.** While query-aware selection is beneficial, simply inserting the selected histories as raw text offers limited gains (*"Raw" in user context*). Instead, constructing profiles from the selected histories (*"Profile" in user context*) provides clearer improvements. By organizing the selected histories into a structured profile, implicit preferences become explicit, and the signals are presented in a more usable form.

**Backbone model naturally influences the overall performance range.** As shown in Table 4, stronger models such as o3-search tend to exhibit consistently higher performance across metrics, whereas relatively weaker models such as GPT-4o-search show a lower performance range. These results indicate that the model's inherent capability establishes the upper and lower bounds of personalization quality. For further analysis of this finding, please refer to Appendix A.2.

**Despite these advances, none of the models in any setting surpass an average score of 60.** This ceiling highlights that personalization in realistic environments remains a significant challenge. In realistic environments, user histories are not clean signals, as preferences are implicit and spread across long-term interactions, which makes them difficult to capture through explicit cues. To advance beyond this limitation, new approaches are needed that can disentangle implicit preferences from noisy interaction data and align them more effectively with the user's query.

## 4.3 FEEDBACK AS A DIAGNOSTIC SIGNAL

Our evaluation framework provides diagnostic feedback, offering a detailed diagnosis of where personalized systems perform well and where they fall short in specific aspects. To demonstrate the diagnostic power of our evaluator's feedback, we analyze two settings discussed in Section 4.2: the configuration exhibiting the lowest personalization performance in GPT-4o-search (w/ full history profile) and the one achieving the most effective personalization (w/ selected history profile).

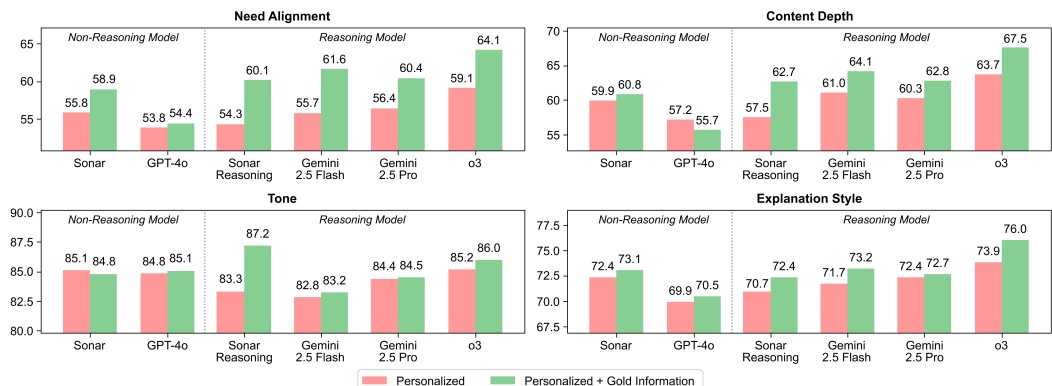

Figure 4: Comparison of personalization performance across models under two settings. *Personalized*(red) generates responses conditioned on user context and the model's own web search results, whereas *Personalized + Gold Information*(green) replaces web search results with gold information.

Figure 3 illustrates the diagnostic feedback generated by our framework for responses of GPT-4o-search to a user query about a balanced diet. In the lowest performance setting, the feedback identifies key shortcomings, such as an overly narrow focus on macros without practical food specifics, basic knowledge depth, and an undesired explanation flow (*red highlights*). In contrast, the highest performance setting effectively addresses these deficiencies, as evident in the response itself (*green highlights*). For these improvements, our evaluation framework explicitly diagnoses how addressed elements are well-integrated through its feedback, while assigning higher scores accordingly to reflect these improvements. By explicitly identifying both strengths and shortcomings, our **feedback for generated response can serve as a diagnostic signal that exposes specific direction for improvement, offering insights for developing more effective personalization methods.**

### 4.4 EFFECTIVE METHOD FOR HISTORY RETRIEVAL

As discussed above, selectively incorporating histories into user context is essential for effective personalization. While leveraging LLMs can directly assess query-history relevance as discussed in Section 4.2, they are prohibitively costly and impractical to scale. Bi-encoder retrievers offer an efficient alternative method for history selection, yet they face fundamental limitations in real-world settings like BESPOKE. Since queries are primarily information-seeking, they often lack sufficient semantic signals to capture the personalization aspects, such as tone or style. Moreover, user preferences are typically implicit and dispersed across long-term interactions, which makes them difficult for bi-encoders to capture using only surface-level semantic similarity (Su et al., 2024). To overcome these limitations, we explore query expansion strategies inspired by prior work (Gao et al., 2023a; Su et al., 2024), including *CoT expansion*, which leverages step-by-step reasoning, and *Pseudo-history expansion*, which synthesizes pseudo-histories to simulate relevant user preferences. More details are provided in Appendix C.4. As shown in Table 5, using the raw query alone yields poor retrieval performance. In contrast, both expansion methods consistently improve performance, demonstrating that **query expansion enriches the query with latent preference cues, making it easier for an efficient bi-encoder retriever to recover relevant histories.**

Table 5: Comparison of history retrieval using original and expanded queries. We use Stella-V5-1.5B model as retriever.

| Query Type | nDCG@10 | nDCG@20 |
|---|---|---|
| Original Query | 0.0820 | 0.1221 |
| +CoT | 0.3809 | 0.3929 |
| +Pseudo-history | 0.3807 | 0.3911 |

### 4.5 IMPACT OF WEB-SEARCH QUALITY ON PERSONALIZED RESPONSES

As discussed in Section 4.2, responses often remain insufficiently personalized even when user context is provided. To pinpoint where this limitation originates, we design an experiment that removes web-search noise by directly providing the model with the gold information while disabling its built-in search tool, thereby ensuring both alignment with the user's information need and the appropriate level of detail. As shown in Figure 4, incorporating the gold information yields consistent improvements in need alignment and content depth across all models. This confirms that the quality of

retrieved information during the web-search stage is a critical factor influencing the personalization quality of the final response. However, the improvements vary considerably depending on the model type. In particular, reasoning models show substantial improvements, while non-reasoning models achieve only modest ones. This indicates that reasoning models are more capable of analyzing and integrating the provided information thoroughly into their outputs.

Additionally, we investigate how the performance gap between reasoning and non-reasoning models changes as the quality of retrieved information deteriorates. To simulate search results under increasing noise conditions, we inject noise by substituting a fraction of the gold information for each query with content drawn from different users and unrelated queries, according to noise ratio. In this experiment, we use Sonar and Sonar-Reasoning, which share the same base model but differ solely in their reasoning capability, enabling a controlled comparison focused on

Table 6: Comparison of personalization performance across different noise ratios for Sonar and Sonar-Reasoning. **Bold** indicates the best scores.

| Model | Noise ratio | Need Align. | Content Depth | Avg. ($\triangle$Avg.) |
|---|---|---|---|---|
| ☀ pplx-sonar | 0% | 58.93 | 60.80 | 59.87 (-) |
| | 30% | 58.80 | 59.87 | 59.33 (-0.90%) |
| | 50% | 57.73 | 57.20 | 57.47 (-4.18%) |
| | 70% | 51.60 | 50.93 | 51.27 (-16.78%) |
| ☀ pplx-sonar -reasoning | 0% | **60.13** | **62.67** | **61.40** (-) |
| | 30% | 60.13 | 60.13 | 60.13 (-2.11%) |
| | 50% | 54.93 | 55.20 | 55.07 (-11.50%) |
| | 70% | 49.47 | 50.27 | 49.87 (-23.13%) |

the effect of reasoning itself. We then compare both models using the two metrics most sensitive to search quality: *Need Alignment* and *Content Depth*. Table 6 shows our experimental results. First, when only gold information is provided, the reasoning model achieves the highest performance. However, as noise increases, reasoning model exhibits a much sharper performance drop compared to the non-reasoning model, and its absolute performance eventually falls below that of the non-reasoning counterpart. This suggests that reasoning models are substantially more sensitive to the quality of the context provided by web search. Taken together, our findings suggest that reasoning models benefit when web search is accurate, but when web search contains noise, they can become disadvantaged relative to non-reasoning models. Therefore, **achieving optimal personalization in information-seeking task requires both accurate web-search and reasoning abilities.**

## 5 CONCLUSION

We introduce BESPOKE, the novel benchmark specifically designed for evaluating personalization in search-augmented LLMs. By constructing a fully human-grounded dataset through long-term annotation, BESPOKE models real-world environments and provides feedback that enables a detailed diagnosis of model strengths and weaknesses. We hope that BESPOKE will contribute to future research on developing personalized systems for more effective information seeking.

### ETHICS STATEMENT

Our work is conducted under the approval of the Institutional Review Board (IRB). During recruitment, annotators are given a clear explanation of the research purpose, procedures, and compensation, and only those who provide informed consent are enrolled. Over the three-week collection period, annotators review their own histories and remove any personally identifiable or sensitive information. We then apply a secondary review and filtering step to ensure robust de-identification and removal of sensitive content. Annotators receive fair compensation for their participation, the adequacy and legality of which is reviewed and confirmed by a legal expert prior to the study. Further details on data collection, anonymization, and compensation procedures are provided in Appendix B.

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

# A DISCUSSION

## A.1 EVALUATING THE EFFECTIVENESS OF OPEN-WEIGHT MODELS AS EVALUATORS

While our GPT-5-based evaluator shows strong empirical performance, this reliance on a proprietary model has two important limitations. It may reduce generalizability across model families, and it incurs non-trivial computational cost for running the benchmark end-to-end. In addition, an evaluation pipeline centered on proprietary systems can be less reproducible and less accessible for the broader research community. Motivated by these limitations, we additionally evaluate whether our evaluator can be replaced by strong open-weight models (GPT-oss-120B and Qwen3-235B-A22B-Thinking-2507). As

Table 7: Meta evaluation result with open-weight models. We use w/ Feedback configuration as $\mathcal{E}_p$

| Model | Criterion | Pearson Corr. | Spearman Corr. | Feedback Acc. |
|---|---|---|---|---|
| GPT-oss 120B | Need Align. | 0.868 | 0.868 | 0.820 |
| | Content depth | 0.827 | 0.831 | 0.787 |
| | Tone | 0.852 | 0.839 | 0.847 |
| | Style | 0.720 | 0.742 | 0.803 |
| | Avg. | 0.817 | 0.820 | 0.814 |
| Qwen3-235B | Need Align. | 0.879 | 0.883 | 0.810 |
| | Content depth | 0.828 | 0.842 | 0.770 |
| | Tone | 0.866 | 0.860 | 0.843 |
| | Style | 0.769 | 0.786 | 0.743 |
| | Avg. | 0.836 | 0.843 | 0.792 |

shown in Table 7, both models produce judgments that closely align with those of human annotators, confirming that effective evaluators can also be built from open-weight models. These results indicate that BESPOKE is not tied to proprietary systems and that its evaluation pipeline can be made more accessible and cost-efficient without sacrificing the reliability of the assessments.

## A.2 ANALYSIS OF BACKBONE MODEL CAPABILITIES FOR PERSONALIZATION

As shown in Table 4, stronger models such as o3-search tend to exhibit consistently higher performance across metrics, whereas relatively weaker models such as GPT-4o-search show a lower performance range. This suggests that the inherent capability of the model sets the upper and lower bounds of achievable personalization quality, and that the effectiveness with which user context is utilized determines how close the model can get to that bound.

It is therefore important to discuss which specific model capabilities are responsible for determining this baseline performance range. Because all models we use are black-box systems, a detailed internal analysis is not possible. However, the most reasonable hypothesis is that both reasoning capability and instruction-following ability play critical roles.

Gemini 2.5 Pro did not exhibit as strong a performance range as we initially expected. This stands in contrast to the performance of o3 on benchmarks such as MMLU-Pro (Wang et al., 2024) and

Humanity's Last Exam (Phan et al., 2025), where the two models show comparable results. While investigating potential causes for this discrepancy, we found consistent evidence that the Gemini family is substantially weaker than o3 on instruction-following benchmarks such as IFEval (Zhou et al., 2023) and IFBench (Pyatkin et al., 2025) (IFEval: 65.4 vs. 95.0; IFBench: 52.3 vs. 69.3).

When using black-box models, all personalization contexts are provided to the model as natural-language instructions. The model should read these instructions and (1) infer the user's information need, (2) retrieve appropriate evidence, and (3) generate a final response that reflects the user's preferred focus, depth, and stylistic preferences. This entire process depends critically on how well the model can follow complex, multi-step constraints specified purely in text.

If a model exhibits weak instruction-following ability, it becomes more likely to misinterpret or under-utilize user-specific cues embedded in the context, leading to degraded personalization throughout both the retrieval and response-generation stages. We believe that this mechanism plausibly explains why backbone models differ so markedly in their achievable performance ranges.

### A.3 DATASET SCALABILITY

While our dataset is relatively small in scale due to its reliance on deeply engaged human annotation, it provides rich user histories that reflect authentic and realistic preferences. These high-quality annotations provide a reliable foundation for personalization research, supplying signals that are difficult to obtain from synthetic or automatically collected data. Building on this foundation, the dataset can serve as seed material for LLM-based augmentation. Grounding augmentation in our dataset enables the synthesis of additional data that is both realistic and user-aligned. This provides a clear path toward scalability, enabling BESPOKE to expand beyond its initial scale while preserving authenticity in richer and more diverse user histories.

### A.4 FEEDBACK AS A REWARD SIGNAL

The feedback in BESPOKE not only enables evaluation but also points to new opportunities for training personalized systems. As discussed in Section A.3, authentic human annotations can be expanded through LLM-based augmentation to construct larger-scale training datasets. Building on such datasets, the feedback has the potential to serve as a rich supervisory signal for developing user-tailored reward models. Unlike scalar ratings, it articulates explicit reasons for user satisfaction or dissatisfaction, which could be directly leveraged in reinforcement learning as a natural language reward signal. This direction suggests a scalable pathway toward reward modeling that remains closely aligned with genuine user preferences, which we leave as future work.

## B DATA COLLECTION DETAILS

### B.1 DETAILS OF RECRUITING HUMAN ANNOTATOR

To collect sufficient user histories, we conduct long-term, deeply engaged human annotation, as detailed in Section 3. Following prior work that leverages long-term human engagement (Kwak et al., 2024; Park et al., 2025b), we collect histories over a three-week period from 30 annotators.

To ensure a diverse and representative benchmark dataset, we recruit human annotators from a wide range of professions and interests, while conducting a thorough screening process to collect valid user histories. We select only those deemed suitable through this screening as annotators. Specifically, we target individuals who are familiar with Google Search and generative AI services such as ChatGPT or Gemini, where "active usage" is defined as engaging with these tools at least 1-2 times per day. Additionally, candidates provide detailed descriptions of their professions and personal interests, allowing us to avoid recruiting individuals similar to those already selected. Instead, we prioritize new annotators with distinct professions and interests to maximize diversity.

As a result, the final annotator cohort spans more than 22 professional domains (including linguistics, pharmacy, law, cybersecurity, medicine, physics, industrial engineering, electrical engineering, marketing, etc.) and over 60 topical interest areas (health, gaming, economics, education, science, finance, literature, cuisine, crime/law, travel, history, arts, AI, statistics, hardware, fashion, environment, etc.). Below is the full set of expertise and interest areas represented in our annotator pool.

**Expertise area:** Linguistics, Pharmaceutical Science, Law, Cybersecurity, AI, Sales Management, Music A&R, Statistics, Medical, Sports Industry, Hardware, Dental, HCI, Physics, Entertainment, Biology, Public Administration, Electronics Engineering, F&B, Industrial Engineering, Materials Engineering, Business Analysis, Marketing

**Interest area:** health, games, economy, performing arts, engineering, science, education, finance, technology, academic papers, news, data science, reading, videos, design, marketing, restaurants/food, document writing, literature, art, delivery, translation, crime, law, security, volunteering, real estate, beauty, business, society, social science, industry, software, shopping, sports, animation, travel, history, foreign language learning, movies, entertainment, cooking, fortune-telling, music, medicine, artificial intelligence, humanities, interior design, automobiles, electronics, politics, religion, alcoholic beverages, startup, employment, computer hardware, computer science, statistics, telecommunications, trends, fashion, programming, academia, academic research, environment

To quantify this diversity, we measure the distribution of annotators' interests using the Shannon index, a widely adopted diversity metric from ecology and information theory. First, we collect detailed interests from each annotator and normalize them into high-level categories $i$ by integrating synonyms and variations (e.g., "yoga, gym workouts" $\rightarrow$ health/fitness), leveraging GPT-5. Next, we aggregate how many annotators' interests include each normalized category. Finally, we compute the relative frequency $p_i$ for each category $i$ and calculate the Shannon index $H$ as:

$$H = -\sum_{i=1}^{S} p_i \log p_i \tag{1}$$

where $S$ is the total number of categories. A higher Shannon index indicates that annotators' interests are spread across diverse categories rather than concentrated in a few. To assess evenness, we compare this value to the theoretical maximum $H_{\max} = \log S$ (achieved when all categories are equally represented) and derive Shannon's equitability $E_H$ as:

$$E_H = \frac{H}{H_{\max}} \tag{2}$$

where $H$ is the observed Shannon index. Our dataset records a Shannon equitability of $E_H = 0.91$, confirming that the annotator group is not biased toward specific topics and instead reflects a broad spectrum of interests, thereby enhancing dataset representativeness.

Each recruited annotator receives a dedicated Google account created for research purposes, with a paid Google AI Pro plan. We also maintain ongoing communication channels between annotators and researchers to promptly address any questions arising during data collection or annotation processes. Prior to participation, all annotators receive detailed documentation outlining the study procedures, compensation structure, data usage policies, and withdrawal terms, and only individuals who provide informed consent are recruited. Annotators receive $50 in monetary compensation, and those who wish to continue using the research account after the study period are granted an additional three months of complimentary AI Pro access. To ensure participant autonomy, annotators are allowed to discontinue their annotation work at any point, whether during or after data collection. In such cases, they receive 50% of the total compensation, and all data contributed by the participant are immediately and permanently deleted. This procedure ensures that annotators are never pressured to continue participation while still receiving fair compensation for their contributions. All compensation schemes and procedures are reviewed in advance by legal experts to ensure compliance with ethical and legal standards.

### B.2 DATA HANDLING & PRIVACY PROTECTION

Our benchmark prioritizes user privacy and ethical data practices through rigorous de-identification process, ensuring no personal information is disclosed without consent. All direct identifiers (e.g., names, emails, phone numbers) are immediately deleted upon collection, while quasi-identifiers (e.g., IP addresses, account IDs) are removed entirely. Location data is generalized to city-level or higher, eliminating finer details. Sensitive expressions in text are filtered using regular expressions and manual review, with problematic content deleted or replaced with neutral alternatives.

Human annotators are fully informed about the de-identification procedures and provided opportunities to review the final anonymized data. To further safeguard privacy, we adopt multi-stage review

processes. The first stage is conducted by annotators themselves via Google Takeout exports, where they can delete sensitive items. The second stage is conducted by our team, who verify and de-identify any remaining identifiable elements through generalization, masking, or removal. Our team manually inspects every session, admitting it into the dataset only when all members unanimously agreed that no re-identification risk remained. In addition to manual inspection, we apply LLM-based screening tools to detect any potential PII, using multiple repeated passes to further minimize risk. Importantly, when we determine that simply masking or modifying certain tokens is insufficient to eliminate potential re-identification risk, we remove the entire session from the dataset.

Only records deemed fully anonymized, free of any risk to annotators or third parties, are incorporated into the dataset. This comprehensive approach guarantees that the user histories provided in our benchmark are ethically sourced, privacy-protected, and pose no risk of unintended disclosure, aligning with the highest standards of data ethics.

### B.3 VALIDATION FOR HUMAN ANNOTATED DATA

**Response-Judgment Pairs.** For the annotation of response–judgment (R–J) pairs, human annotators are asked to evaluate model responses along the four personalization criteria: need alignment, content depth, tone, and explanation style. To ensure high-quality and consistent annotations, we provide detailed guidelines with illustrative examples for each criterion, allowing annotators to clearly understand the evaluation standards. During the process, annotators review their own feedback three times to check whether their judgments are consistent and well aligned with their preference. This review process ensures that the collected feedback is both reliable and consistent for each annotator themselves, resulting in trustworthy R–J pairs for evaluating personalization.

**Gold Information Set.** For each query, we construct the candidate gold information set by leveraging GPT-5 (*reasoning effort = high*) with the prompt outlined in Table 14, to extract atomic claims from the gold response $r^+$, following methods similar to RAGChecker (Ru et al., 2024) and FActScore (Min et al., 2023). For improving accuracy and consistency, we manually annotate gold information for two sampled instances and incorporate these annotations as exemplars in a 2-shot prompting strategy. Subsequently, we perform a manual inspection of this candidate set to ensure its appropriateness and quality. Since the information is derived directly from the response, it may contain personally identifiable elements, which are promptly removed to ensure privacy and uphold ethical standards, as mentioned in Appendix B.2. Next, we check whether each claim can be verified through real-world web sources, retaining only those that can be verified through Google Search. Finally, to eliminate redundancy, we merge claims that convey identical information but differ only in minor phrasing, resulting in a gold information set $\mathcal{I}_q^+ = \{i_{q,1}^+, \ldots, i_{q,n}^+\}$.

### B.4 INSTRUCTION FOR HISTORY COLLECTION

Annotators are instructed to record search and chat histories exactly as they normally would, without prompting them toward specific topics or styles. Minimal intervention is used to preserve naturalistic behavior. We also provide examples of typical everyday information-seeking behaviors, ensuring the collected histories reflect real-world user signals suitable for personalization evaluation.

### B.5 INSTRUCTION FOR QUERY ANNOTATION

Annotators are given four predefined information-seeking tasks discussed in Section 3.3, along with multiple example queries for each task. They are instructed to phrase queries naturally, just as they would when using ChatGPT or Gemini. Each query also required a gold information need, which annotators were asked to describe as explicitly and concretely as possible.

### B.6 INSTRUCTION FOR RESPONSE-JUDGMENT PAIRS ANNOTATION

For each query, annotators are given six responses (one gold response + five randomly sampled responses). For each response, annotators provided a scalar rating, and a free-form diagnostic ex-

planation. Explanation of each rubric is shown for every annotation page to ensure consistent interpretation and reduce subjectivity.

### B.7 ANNOTATION QUALITY CONTROL

We implement several layers of quality assurance throughout the annotation pipeline. During the query annotation stage, any query that is inconsistent with the task definitions is sent back for revision, and gold information needs are manually inspected and refined when they lacked sufficient specificity. During the judgment stage, feedback entries that are overly short or missing specific feedback triggered revision requests, requiring annotators to re-submit improved annotations.

## C EXPERIMENT DETAILS

### C.1 OVERALL

We evaluate six representative search-augmented LLMs: GPT-4o-search, o3-search, Gemini-2.5-Flash, Gemini-2.5-Pro, Perplexity-sonar, and Perplexity-sonar-reasoning. The prompts used for inference with these models are provided in Tables 15 and 16.

To measure recall for gold information coverage, for each query, we iterate through the gold information set and use GPT-5 (*reasoning effort = high*) to perform binary classification on whether each individual gold information item is included in the response without contradiction. The recall is then computed as the proportion of gold information items that are correctly included. The prompt used for this classification is shown in Table 17. For personalization evaluation, we provide the query $q$, the gold information need $n_q^+$, the R-J pair for the query, and the corresponding gold rubric $\mathcal{G}^+$ as evaluation context. Based on this, we generate scores $s$ and feedback $f$ for the response across the four criteria with GPT-5 (*reasoning effort = high*). The prompt for generating the gold rubric is provided in Table 18, and the prompt used for evaluation is shown in Table 19.

### C.2 META EVALUATION

In the meta-evaluation, we compare our personalized LLM-based evaluator $\mathcal{E}_p$ against baselines across different settings. The non-personalized version (w/o Personal.) is implemented by providing only the query, response, and general definitions of the individual criteria as the evaluation context. The score-only baseline (w/o Feedback) incorporates the query, the gold information need $n_q^+$, the response, R-J pairs containing only scores, and a gold rubric is generated from these score-only R-J pairs and included in the evaluation context. Our full evaluator (w/ Feedback) follows the approach described in Section 3.4, utilizing the complete evaluation context including feedback information from the R-J pairs to assess the responses. For the feedback alignment assessment, the prompt used with GPT-5 as the meta-evaluator is provided in Table 20. This prompt instructs the meta-evaluator to determine if the evaluator-generated feedback and human-annotated feedback convey equivalent content, given the context $(q, n_q^+, \mathcal{R}_q^+)$, and outputs a binary label for equivalence.

### C.3 USER CONTEXT ANALYSIS

In our experiments, all user profiles were generated using GPT-5 (*reasoning effort = high*). For the static profile, we aggregated the complete history of each user into a single profile, with the prompt used for this generation shown in Table 21. For history selection, we leverage GPT-5 in a cross-encoder manner, following methods similar to Niu et al. (2024) and Weller et al. (2025), to evaluate the relevance of each history session to a given query. The relevance was defined as whether the history would contribute to generating a personalized response, and the judgment was made in a binary classification. Only histories deemed relevant were retained. The prompt for this relevance assessment is provided in Table 22. For the dynamic profile, we constructed an adaptive profile for each query. We considered two variants: one based on the entire history, and another restricted to the subset of histories identified as relevant (indicated by ✓ under History selection). The prompt for dynamic profile generation is presented in Table 23.

Table 8: An example where the same query results in different information needs depending on the user, showing how User19 (AI researcher) emphasizes technical mechanisms and the role of artificial intelligence, while User20 (media producer) highlights sociocultural aspects.

| **Query:** How has the spread of social media affected people's lives? | |
|---|---|
| *User19*: Works in AI research and development, with a **strong interest in the latest AI technologies**. | *User20*: A media-related PD with a strong **interest in social phenomena and cultural trends**. |
| *Gold information need:* I'm curious about the **influence of social media in the age of AI**. Could you explain, with specific examples, how it has affected various aspects of our daily lives? | *Gold information need:* Analyze how the spread of social media has impacted the lives of modern people. In particular, analyze changes in **family relationships, the emergence of new professions** ... |
| *Gold Response:* **Recommender systems** rank posts by signals like watch time, likes, comments, replays, and pauses (dwell time) ... A "filter bubble" is when this personalization narrows what you see ... **Generative AI** creates text, images, audio, and video from prompts ... **Machine translation** ... | *Gold Response:* Social media lowered the cost of connection and raised the visibility of social proof, reshaping **family relationships, job structures**, and daily routines ... **Families:** group chats and perpetual photo/video sharing strengthen bonds across distance ... **Creators/influencers/streamers** produce ... |

## C.4 HISTORY RETRIEVAL

To evaluate the effectiveness of history retrieval methods, we compare the performance of using the raw query alone against query expansion techniques. Specifically, we consider two variants of query expansion to enhance history retrieval: CoT expansion (Su et al., 2024), which prompts the model to infer the key aspects of a raw query most important for retrieving relevant past sessions through step-by-step reasoning, and pseudo-history expansion, which adapts the HyDE (Gao et al., 2023a) idea of pseudo documents by generating a pseudo history that serves as a synthetic representation of the personalization signals likely to be relevant for retrieval. The prompts employed for these expansion methods are detailed in Table 25.

To measure history retrieval performance, we designate oracle histories using an LLM as ground truth and evaluate the retriever's performance. Specifically, for each query, we first establish an oracle set of histories by leveraging the gold rubric $R_q^+$, which captures the ideal personalization aspects. Using GPT-5, we iterate over all sessions in the user's interaction history, assessing for each whether it contains, query-specific personalization signals that align with the gold rubric—such as stable preferences, explicit constraints, background context, or adjacent decision-changing factors—and designate those that do as the ground-truth oracle histories. The prompt employed for oracle history selection is presented in Table 24. Subsequently, we assess the retrieval quality by inputting the actual query (or its expanded version) into the bi-encoder retriever and measuring how well it recovers these oracle histories, measured via nDCG@K.

## C.5 DATA CONSTRUCTION & COST BREAKDOWN

For generating hypothetical information needs, we set $k = 6$. The overall annotation pipeline incurs both human and LLM-assisted costs. Each human annotator is compensated $50 in total ($25 for constructing user histories and $25 for producing query annotations and judgments.) For LLM-assisted annotation using GPT-5 (with high reasoning effort), the cost is $0.015 per query for gold rubric generation and $0.005 per query for gold information-need generation, resulting in an average annotation cost of approximately $0.35 per query. For evaluation, our GPT-5–based evaluator adds an additional $0.05 for personalization-quality scoring and $0.05 for gold information coverage scoring, based on evaluating each model response against roughly 20 gold claims.

## D CASE STUDY

Personalization is essential in information-seeking tasks because the same query can correspond to different information needs across users. For qualitative evaluation, we conducted a case study in the BESPOKE dataset focusing on instances where users with different backgrounds issued the same test query. As shown in Table 8, User 19, an AI researcher, consistently focuses on technical mechanisms and AI-mediated effects, whereas User 20, a media producer, emphasizes sociocultural phenomena and cultural trends. For the query *"How has the spread of social media affected people's lives?"*, User 19's gold response emphasizes keywords such as recommender systems, generative AI, and

Table 9: Full statistics of BESPOKE

| User ID | #Search Sessions | #Chat Sessions | Avg. turns per Chat Session | Max turns per Chat Session |
|---|---|---|---|---|
| 1 | 108 | 49 | 4.65 | 10 |
| 2 | 11 | 18 | 4.50 | 12 |
| 3 | 61 | 26 | 4.85 | 18 |
| 4 | 63 | 19 | 8.58 | 53 |
| 5 | 175 | 36 | 2.67 | 15 |
| 6 | 24 | 19 | 5.16 | 13 |
| 7 | 192 | 18 | 4.67 | 21 |
| 8 | 40 | 32 | 4.56 | 18 |
| 9 | 83 | 12 | 3.42 | 10 |
| 10 | 47 | 25 | 1.88 | 4 |
| 11 | 51 | 24 | 2.71 | 7 |
| 12 | 6 | 27 | 5.19 | 12 |
| 13 | 61 | 52 | 4.58 | 20 |
| 14 | 28 | 12 | 4.25 | 19 |
| 15 | 238 | 38 | 3.00 | 11 |
| 16 | 58 | 13 | 3.31 | 7 |
| 17 | 67 | 51 | 4.53 | 25 |
| 18 | 55 | 36 | 2.31 | 10 |
| 19 | 68 | 38 | 3.92 | 10 |
| 20 | 56 | 11 | 7.27 | 31 |
| 21 | 113 | 9 | 4.22 | 8 |
| 22 | 73 | 28 | 2.43 | 10 |
| 23 | 15 | 8 | 4.25 | 9 |
| 24 | 73 | 12 | 6.00 | 21 |
| 25 | 32 | 21 | 4.05 | 20 |
| 26 | 16 | 17 | 3.24 | 8 |
| 27 | 62 | 21 | 4.57 | 17 |
| 28 | 202 | 22 | 6.45 | 16 |
| 29 | 53 | 12 | 3.42 | 7 |
| 30 | 22 | 11 | 4.36 | 8 |
| **Avg.** | 73.48 | 24.34 | 4.30 | 15 |

machine translation, while User 20's gold response highlights keywords such as family relationships and job structures. This divergence underscores that an identical query does not uniquely determine the information need; it depends on the user's background and preferences. Consequently, a truly effective personalized search-augmented LLM must accurately infer the user-specific needs behind the query. To achieve this, the model could leverage user history as a key signal for preference inference, ensuring that the generated responses are appropriately tailored to the individual.

# E  DATASET STATISTICS

Table 9 summarizes the overall statistics of BESPOKE. On average, each user contributes 73.5 search sessions and 24.3 chat sessions, indicating that the dataset balances both query-driven and conversational interactions. Search sessions reflect realistic information-seeking behavior, while chat sessions capture interactive exchanges that vary in length and style. Chat sessions contain 4.3 turns on average, with some sessions extending up to 53 turns, demonstrating substantial variation in conversational depth. In total, we observed 2,983 chat turns across all users. Combined with the 2,153 search sessions in the search history, BESPOKE contains 5,136 interaction history overall. This distribution highlights that BESPOKE captures both short, focused dialogues and long, exploratory conversations, providing a balanced benchmark for evaluating personalized search-augmented LLM across heterogeneous user behaviors.

# F  DATA EXAMPLES

## F.1  GOLD INFORMATION NEED AND R-J PAIRS

In Table 10, we present an example of a user query, the corresponding gold information need, and multiple R-J pairs. The query and gold information need are written by human annotators to ex-

plicitly capture the user's true intent, while the paired responses show how the user assigns different scores and feedback to different model outputs.

### F.2 User history and gold rubric

Table 11 illustrates samples of the user's chat history session and search history session, demonstrating how past interactions and browsing activities help shape the interpretation of the current query. The chat history consists of multi-turn user-assistant conversations, while the search history contains user queries along with visited site titles and page summaries. The table also introduces the gold rubric we used for evaluation, which defines four dimensions: need alignment, content depth, tone, and explanation style, with standardized 1–5 scoring guidelines. This rubric ensures consistent and interpretable personalized judgments while providing diagnostic signals of model behavior.

### F.3 Comparison of R-J pairs in different history utilization setting

To further illustrate the effect of different history utilization settings, we provide two extended examples from Table 12 (Perplexity-sonar) and Table 13 (Gemini-2.5-flash). In both tables, the upper example corresponds to the responses generated with a full history profile, while the lower example corresponds to the responses generated with a query-aware selected history, holding all other experimental settings constant. These comparisons show how diagnostic feedback exposes concrete weaknesses (red) and improvements (blue) that arise from different user-context constructions.

In Table 12, the upper example reveals several shortcomings. Although the destinations satisfied the Romance-language and regional preferences, the model did not explicitly map each recommendation to the five specified constraints: budget, safety, English usability, car-free mobility, and trip length. The response read as high-level advice, omitting field-ready details such as transit pass names, concrete price ranges, and concise day-by-day itineraries. Moreover, the tone drifted into a marketing-like style, making assumptions about the user's professional identity, which the annotator flagged as inappropriate. By contrast, the lower example addressed some of these gaps by providing clearer per-destination budgets and structured itineraries. The diagnostic feedback recognized this improvement in both need alignment and explanation style, while still urging the inclusion of explicit safety and mobility mapping and stricter adherence to a standardized template.

In Table 13, a similar pattern emerges. In the upper example, the model produced a broad overview centered on brands and communication, but failed to analyze the three facets explicitly requested: family relationships, the emergence and decline of professions, and lifestyle shifts. The feedback noted that the response relied on generic claims with no selective data and no concrete examples, leaving content depth and need alignment weak despite a neutral tone. With the lower example, the output moved closer to the intended report format by explicitly covering family, jobs, and lifestyle and grounding them in platform-specific mechanisms. Nevertheless, the feedback emphasized that the analysis still lacked quantitative evidence, explicit mention of declining professions, and the structured reporting style that was requested. Taken together, these cases show that while utilizing query-aware selective histories consistently improves response quality, effective personalization also requires systematic coverage of all 4 criteria.

## G Usage of Large Language Models

For this work, we use an LLM solely to assist writing with minor grammatical issues. We do not receive any assistance regarding the overall structure or the ideas of the paper.

## H Prompts

Table 10: Example of gold information need and two Response–Judgment (R-J) pairs, which are discussed in Section 3.3

| **Query** |
| --- |
| What is digital literacy and why is it important? |

| **Gold information need** |
| --- |
| The user wants a clear definition of digital literacy, a breakdown of its core skills/components, examples of how it is applied in practice, and recent extensions such as AI literacy. They also want synonyms/antonyms (e.g., media literacy, information literacy vs. digital illiteracy) and a diachronic perspective (how the concept has changed over time). |

| **Sampled response 1** |
| --- |

**Digital Literacy, A Deeper Dive!**
Digital literacy goes beyond simply using digital devices. It's the integrated ability to find the information you need amid the flood of digital data (Access), critically analyze its authenticity (Analyze), create meaningful content of your own (Create), and communicate and act responsibly with others (Action). Think of it as having a sturdy ship and a precise compass to navigate the digital world! ...

**What Makes Up Digital Literacy? (Detailed Explanation!)**
- **Technical Proficiency**: more than just using tools, but understanding how they work and combining them to suit your needs. - **Critical Thinking & Curation**: identify who created information, purpose, and bias. - **Creative Production**: use digital tools to turn ideas into content...

**Practical Examples by Field — When Is It Needed?**

| Field | Practical Example | |
| --- | --- | --- |
| Education | History class with VR ruins; Science class analyzing air pollution data with Excel | |
| Workplace | Marketing campaigns with data-driven targeting; workflow automation with RPA/Python | ... |
| Daily Life | Open banking dashboards; fact-checking fake news with SNU FactCheck Center | |

| **Human judgment 1** |
| --- |
| **Need Alignment: 4/5**, It covers most of what I wanted: clear definition, skills/components, where it's used, and timely AI-era updates with practical examples. But it skips the synonyms/antonyms I asked for, and a quick then-vs-now line would make the diachronic angle explicit.
**Content Depth: 5/5**, Totally solid right now: concrete, real-world examples and current tools make it genuinely useful without fluff.
**Tone: 4/5**, Generally pleasant and friendly, but the emoji and exclamation points feel a bit heavy for me, lighter would read more naturally.
**Explanation Style: 4/5**, Clear sections and examples, but Practical Examples by Field table is hard to read; missing "Synonyms/Antonyms" subheading. |

| **Sampled response 2** |
| --- |

**Digital Literacy: An Essential Competency in Modern Society — Importance and the Digital Divide**
Digital literacy is a core competency for individuals and society. Beyond technical skills, it encompasses critical understanding of information, communication, and value creation ...

**The Multidimensional Importance and Positive Impact of Digital Literacy**
- Expanding Employment and Economic Opportunities — demand in AI, big data, digital marketing; platforms enabling entrepreneurship. - Innovation in Education — online platforms, VR/AR for personalized learning; foundation for lifelong learning. - Strengthening Democratic Participation — digital media as a forum, discern misinformation, promote digital citizenship. ...

**The Digital Divide: Deepening Inequality**
- From first-level (access) → second-level (skills) → third-level (outcomes). - Worsening income inequality due to automation/AI. - Political exclusion and crisis of representation for vulnerable groups. ...
In conclusion, digital literacy is no longer optional but a vital survival skill. ...

| **Human judgment 2** |
| --- |
| **Need Alignment: 2/5**, I wasn't asking about the digital literacy gap. You didn't directly cover what I asked for: a clear definition, the key skills/components, where it's used, synonyms/antonyms, and how it's changed over time. Most of the space fixates on the digital divide instead of those four facets.
**Content Depth: 2/5**, The quality of the content itself isn't bad, but it's mostly high-level prose. I wanted concise, concrete skills and examples. Instead I got big-picture claims with few actionable details, so it reads like a mini-paper more than a useful, specific answer.
**Tone: 2/5**, It's too stiff and boring. It feels full of intellectual vanity. I don't like seeing you use such pretentious language—even the title reads like a conference paper.
**Explanation Style: 3/5**, It was good that you divided it into two sections, each with its own subheading. But it doesn't map to the facets I asked for, and the paragraphs feel heavy. Please switch to a modular layout with short bullets under those headings. |

Table 11: Example of user history and gold rubric for scoring Response–Judgment pairs

| Query |
| --- |
| What is digital literacy and why is it important? |

| Chat history session |
| --- |
| **User**: Please recommend some trending web novels lately, but leave out romance and romance fantasy. |
| **Assistant**: You're looking for some trending web novels, huh? Leaving out romance and romance fantasy, I've picked out only the most exciting stories for you. Lately, genres where the main character grows, overcomes adversity, gets satisfying revenge, or shines as a professional are popular. - The Scoundrel Has Returned (Fantasy): synopsis, why it's popular, where to read... |
| **User**: I told you to bring me links. |
| **Assistant**: Apologies for missing the links. Here are versions with links... |

| Search history session |
| --- |
| **Search query**: AI webtoon and web novel market |
| **Visits**: <Title> Matured Webtoons and Web Novels: Breaking Genre Boundaries and Offering Personalized AI Recommendations |
| <Page summary> The page discusses the strategy in responding to the maturing webtoon and web novel market. As the market enters a new phase, the company plans to diversify genres and expand user engagement, incorporating intellectual property from games, films, and YouTube. They are also leveraging advanced AI technologies to analyze user preferences and offer personalized content recommendations while breaking traditional genre boundaries... |

| Gold rubric |
| --- |
| **Need Alignment** |
| - 5: Directly answers the exact facets the user requested, in this order or equivalent: definition; skills/components; fields/situations of use; synonyms/antonyms; diachronic change (then vs now). Incorporates up-to-date elements (AI, deepfakes, algorithmic feeds/bias, IoT). No detours or filler. |
| - 4: Covers nearly all facets with only a minor omission or light coverage (e.g., brief or missing synonyms/antonyms), and minimal drift. |
| - 3: Generic treatment; misses one or two key facets or treats them superficially; modern angles underemphasized. |
| - 2: On-topic but focuses on secondary angles (e.g., long digital-divide discussion) or analogies instead of requested facets; multiple facets missing. |
| - 1: Largely off-target or ignores the requested facets. |
| |
| **Content Depth** |
| - 5: Concise yet substantive; mixes clear concepts with actionable, field-specific examples and named tools/workflows; basic vs advanced snapshots when helpful; reflects current realities (AI verification, deepfakes, algorithmic bias, passkeys/MFA); no fluff. |
| - 4: Solid detail with a few gaps; mostly actionable; minor depth or recency improvements needed. |
| - 3: Understandable but high-level or generic; lacks concrete tools/cases or field specificity; user must infer steps. |
| - 2: Too abstract/academic or too shallow; few/no named tools or workflows; limited practical value. |
| - 1: Severely mismatched depth (mini-paper or basic listicle) that the user cannot use. |
| |
| **Tone** |
| - 5: Calm, professional, friendly; natural phrasing; at most 1–2 emojis/exclamations; no hype, cutesiness, pretension. |
| - 4: Generally aligned; slightly dry or a few extra emojis/exclamations, but not distracting. |
| - 3: Neutral/robotic or somewhat academic; acceptable but not preferred. |
| - 2: Off-putting: forced cheerfulness, heavy emojis/exclamations, cutesy metaphors, or stiff/pretentious vibe. |
| - 1: Condescending, arrogant, or offensive. |
| |
| **Explanation Style** |
| - 5: Clean, modular structure mapping to facets: Definition; Skills/Components; Where Used; Synonyms/Antonyms; Diachronic Change. Short bullets/brief paragraphs; separates concepts from examples (e.g., basic vs advanced); optional key takeaway; avoids dense tables. |
| - 4: Clear sections and mostly bullets; minor misalignment (order or one missing subheading), slightly wordy, or a small but readable table. |
| - 3: Some structure but heavier paragraphs or headings that don't map cleanly; requires extra effort to scan. |
| - 2: Loose/essay-like; long blocks; structure not matched to requested angles; dense, hard-to-read tables. |
| - 1: Disorganized; no clear sections; incompatible format for quick scanning. |

Table 12: Comparison of Perplexity-sonar response judgments in a different history retrieval setting

| **Query** |
| --- |
| Please recommend some overseas travel destinations. |

| **Gold information need** |
| --- |
| I want to travel somewhere for 5 days with a budget of under USD 3,000. Here are my criteria: 1) I'm interested in Romance language cultures, 2) I want to travel to Europe or the Americas, 3) The destination should be reasonably safe, 4) I should be able to communicate at least partially in English, and 5) I need to be able to get around without a car. Please recommend countries and regions that fit these requirements. |

| **Perplexity-sonar response with full history profile** |
| --- |
| Based on your interest in practical AI workflows, Gen Z communications, and travel in Europe/South America, here are some **personalized overseas travel destinations** that fit well with efficient planning, ethos-aware choices, and engaging storytelling for your audience or career-related interests: 
 **1. Spain (Barcelona and Madrid)** - Easily accessible by low-cost airlines and extensive train networks (Renfe) ideal for efficient, budget-smart travel. - Vibrant Gen Z culture hubs for marketing storytelling and CSR-related event activations. - Plenty of English usability in major urban centers, plus rich options for sustainable travel choices (walking tours, urban bike rentals). - Use Rome2Rio and Skyscanner for streamlined booking and cost comparison between flights and trains. - Safety-wise, standard European precautions apply, ... |

| **Evaluator judgment** |
| --- |
| **Need Alignment: 2/5**, The picks mostly fit Romance-language Europe/Americas, but you didn't map each destination to my five conditions or show that a 5-day trip fits under USD 3,000. Please add, per destination: safety, English usability, and car-free mobility (explicitly), plus a budget breakdown (flight, lodging/night, daily spend, 5-day total). Also, avoid assumptions about me (e.g., Gen Z/CSR focus). 
 **Content Depth: 2/5**, This reads as high-level advice. I need field-ready details: name the transit cards/passes and how to get around without a car, concrete price ranges, a few local neighborhoods/hidden gems, a concise one-day walking plan, practical tips (e.g., pickpocket hotspots, booking/free hours), and sensible day-trip ideas. 
 **Tone: 2/5**, The framing assumes my professional interests (Gen Z, CSR) and feels like a marketing pitch. Please keep it calm and straightforward, lead with the bottom line, and be transparent with ranges and trade-offs—without assumptions about who I am. 
 **Explanation Style: 3/5**, You use bullets, but it's not in a consistent, scannable template and lacks costs. Please structure each destination as: Highlights; Fit to my 5 conditions; Getting around (no car, pass names); Estimated costs (flight, lodging/night, daily, 5-day total); Local favorites; Sample one-day itinerary; Practical tips. |

| **Perplexity-sonar response with query-aware selected history profile** |
| --- |
| Here is a personalized shortlist of overseas travel destinations tailored to your preferences—Romance-language countries, good English usability in tourist areas, typical trip length of 5–7 days, and a total budget target around USD 3,000 for the entire trip: 
 **Western Europe** 
 **1. Spain (Barcelona + Day Trips to Girona or Montserrat)** 
 **Why fit:** Spanish is Romance language; Barcelona is vibrant with good English in tourist zones. You've also researched Spain's one-week budget already, so you're familiar. 
 **English usability:** High in hotels, tourist restaurants, museums, public transport signage mostly bilingual. Some local shops less so. 
 **5-day idea:** Base in Barcelona; day trip to Girona or Montserrat Monastery. 
 **7-day idea:** Add a second base day or another day trip to Sitges or Tarragona... |

| **Evaluator judgment** |
| --- |
| **Need Alignment: 4/5**, Good picks (Romance-language, Europe/Americas) and solid cost breakdowns with 5-day totals. However, you didn't explicitly map safety and car-free mobility for each destination, ... please avoid assumptions like "You've already researched Spain's one-week budget." 
 **Content Depth: 3/5**, Useful budgets and quick day-trip ideas, but it's still generic. Please add field-ready details: specific transit passes with prices (e.g., Barcelona Hola BCN/T-Casual; Lisbon Viva Viagem; Porto Andante; Paris Navigo Easy/Paris Visite; Rome Roma Pass), a few local neighborhoods/hidden gems (e.g., Gràcia/El Born; Alfama/Mouraria; Canal St-Martin; Trastevere), a concise one-day walking plan per city, and practical tips (pickpocket hotspots, free museum hours, booking windows). 
 **Tone: 4/5**, Clear and calm overall. One assumption about my prior Spain research and no upfront bottom line. Lead with the conclusion (e.g., "Bottom line: Portugal and Spain comfortably under $3000; France/Italy are tight; South America not advised for 5 days") and keep avoiding assumptions. 
 **Explanation Style: 4/5**, Nicely bulleted and mostly consistent, but not in the preferred per-destination template. Please standardize: Highlights; Fit to your 5 conditions; Getting around (no car) with pass names; Estimated costs (flight, lodging/night, daily, 5-day total); Local favorites; Sample one-day itinerary; Practical tips. Put South America under a clear "Why not for a 5-day trip" section. |

Table 13: Comparison of Gemini-2.5 response judgments in a different history retrieval setting

| Query |
| --- |
| How has the spread of social media affected people's lives? |

| Gold information need |
| --- |
| Analyze how the spread of social media has impacted the lives of modern people. In particular, write a report analyzing changes in family relationships, the emergence of new professions and the decline of others, and shifts in lifestyle both online and offline, using examples. |

| Gemini-2.5-flash response with full history profile |
| --- |
| Social media has profoundly reshaped the landscape of human interaction, communication, and commerce, creating both significant opportunities and considerable challenges in people's daily lives. From revolutionizing how content is created and distributed to transforming brand engagement and fostering new avenues for community, its influence is pervasive. However, this digital ubiquity also brings concerns regarding mental well-being and the spread of misinformation. Here's a breakdown of how social media has affected various aspects of life: **Decision Matrix:** ... |

| Evaluator judgment |
| --- |
| **Need Alignment: 2/5**, This centers on creators/brands/AI and general communication; it doesn't analyze family relationships, the emergence/decline of professions, or online/offline lifestyle shifts with concrete examples. Please reframe into those sections and tie each point to social-media mechanisms with platform-specific examples.
**Content Depth: 2/5** Mostly broad, well-known points with minimal mechanisms, no selective data, and few concrete cases. Add 2–5 data points, clear causal reasoning, and specific examples per section (e.g., family vlogging on YouTube/TikTok; new roles like UGC creators/community managers vs. declines in print ad sales roles), plus a short near-term outlook.
**Tone: 4/5**, Neutral and readable overall. It feels a bit like a marketing deck and somewhat wordy—tighter, more concise phrasing would improve it.
**Explanation Style: 3/5** The matrix and checklist help, but the structure doesn't match the requested report. Organize into Family / Professions (new & declining) / Lifestyle (online & offline) with bullet points, selective data and platform examples, plus a brief executive summary and outlook. |

| Gemini-2.5-flash response with query-aware selected history profile |
| --- |
| The pervasive spread of social media has fundamentally reshaped how people connect, work, and live, presenting both remarkable opportunities for connection and significant challenges to well-being and productivity. Its net effect is a complex tapestry of enhanced access, fragmented attention, and shifting social norms, demanding conscious navigation of its platforms and pervasive influence. Here's an at-a-glance map of key impact domains: - **Jobs & Work:** New career paths, personal branding, algorithmic work. - **Lifestyle & Time Use:** Attention fragmentation, habit formation, self-presentation. - **Family Relationships:** Intergenerational dynamics, communication shifts, privacy. - **Information Ecosystem:** News consumption, misinformation, polarization. - **Mental Health:** Social comparison, anxiety, loneliness... |

| Evaluator judgment |
| --- |
| **Need Alignment: 4/5**, You addressed family, jobs, and lifestyle with platform-specific examples, but you didn't clearly cover the decline of professions and you drifted into secondary domains (news, civic life, education). Please focus the report on the three facets, call out specific declining roles due to social media, and make offline lifestyle shifts (e.g., shopping, dining, travel behaviors) explicit.
**Content Depth: 3/5**, The mechanisms help, but there are no selective data points, named cases, or near-term predictions. Add 2–4 credible stats per section (e.g., ad spend shifting to social, creator economy size, % of families using group chats, live-commerce penetration), concrete cases, clearer causal links (why social media causes job declines), and short forecasts tied to each facet.
**Tone: 4/5**, Clear and neutral overall; a bit wordy and slightly conversational in places. Tighter phrasing and sticking to concise, objective language would sharpen it.
**Explanation Style: 4/5**, Headings/bullets make it skimmable, but it's not in the exact report format. Mirror the requested structure—Family / Professions (new vs declining) / Lifestyle (online vs offline)—add a brief executive summary and near-term outlook, and include bullet-point examples plus 1–2 data points per section; keep secondary impacts brief or label them as an add-on. |

Table 14: The prompt for extracting candidate gold information set

**Prompt for extracting candidate gold information set**

You are an information extraction assistant. Given a user query and a model response, extract atomic information claim that represent 'gold information' aspects strictly following these rules.
General rules (apply in all cases):

Output strictly JSON with a single top-level key: gold information (array of strings). No other keys or text.
Each item should be a single atomic information claim describing an aspect.
Exclude any time-related content.
Exclude user-specific private attributes or unverifiable personal details.
Do not include proper nouns (titles/brand names/person names); use common-noun categories instead.
If price information appears, collapse multiple prices into a single range or ceiling/floor statement.

Recommendation-style queries (when the response recommends items):
Extract aspects of the recommended items without their specific names.
Use the common-noun form for the item (e.g., 'Game', 'Headphones', 'Laptop').

Information-seeking queries (all other cases):
Extract factual information presented in the response, following all general exclusions above.

<Query>: {query}
<Response>: {gold response}
Return JSON only in the form: { "gold information": [ ... ]}

Table 15: The prompt for non-personalized response generation

**Prompt for non-personalized response generation**

You are a personalized information-seeking assistant.
<Query>: {query}

Provide a personalized answer based on user context and accurate web-searched information. Do not ask follow-up questions or provide generic responses - deliver a direct, tailored answer to the user's query.

Table 16: The prompt for response generation with user context

**Prompt for personalized response generation with user context**

You are a personalized information-seeking assistant.
<Query>: {query}
<User Context>(follow these preferences to craft the answer): {user context}

<GUIDELINES>
- Match the organization, tone, depth, and style implied by the user context
- Provide a personalized answer tailored to the user's preferences through web search.
- Use accurate, up-to-date information obtained through web browsing.
- Do not ask follow-up questions or provide evaluations; output only the final personalized answer.

Provide a personalized answer based on user context and accurate web-searched information. Do not ask follow-up questions or provide generic responses - deliver a direct, tailored answer to the user's query.

Table 17: The prompt for $\mathcal{E}_r$

**Prompt for $\mathcal{E}_r$**

You are a precise fact checker. Decide if the response contains the claim's core meaning. Answer strictly with true or false. Consider it contained if semantically equivalent even with different words. Do not count if contradicted or absent.

Claim: {claim}
Response: {response text}
Return strictly one token: true or false.

Table 18: The prompt for generating gold rubric based on R-J pairs

**Prompt for generating gold rubric based on R-J pairs**

You are an expert annotator who derives a single user's personalized evaluation rubric from their past evaluations.
Analyze responses in the JSON and infer the user's implicit standards.
Output only the analysis text between <BEGIN ANALYSIS> and <END ANALYSIS>
{R-J pairs for each query}

These evaluations of the query and gold information responses were all done by one person.
For each response, they evaluated 4 criteria on a 1-5 scale with feedback:
Base scale guidance (1-5 for each criterion):

Need Interest Alignment:
1: Irrelevant to the user's actual interests within the broader topic.
2: On-topic but mainly secondary details the user likely finds uninteresting.
3: Generic one-size-fits-all; fails to reflect the user's specific interests.
4: Captures core interests with minor drift toward less important details.
5: Pinpoints the exact core concepts and perspectives the user is most curious about.

Content Level:
1: Completely mismatched depth (e.g., research paper for a simple query).
2: Too hard to comprehend or so basic it has very little value.
3: Understandable but noticeably mismatched; requires extra effort or context.
4: Overall a good match; could be slightly deeper or simpler to be perfect.
5: Perfectly matches the user's knowledge level; effortless to comprehend.

Tone:
1: Very uncomfortable or offensive to the user.
2: Disliked tone is prominent, causing discomfort/resistance.
3: Functionally okay but robotic or distant from preference.
4: Generally close to preferred tone with a few awkward expressions.
5: Perfectly emulates the user's preferred tone and manner; exceptionally natural.

Explanation Style:
1: Completely incompatible with the user's way of learning.
2: Structure/style does not fit, making it hard to absorb.
3: Understandable but requires more effort than preferred.
4: Mostly preferred and easy to follow; minor clarity improvements possible.
5: Exactly the methods/expressions easiest for the user (e.g., analogies, summaries, bullets).

Based on the rubric above, review how the user evaluated responses for the query and derive the user's personalized evaluation rubric.
Provide your result strictly between <BEGIN ANALYSIS> and <END ANALYSIS> in the format below:
- Provide a concise, actionable personalized Rubric for the user for each criterion. You SHOULD provide all 5 scores for each criterion. - > (1) Personalized rubric with 1–5 scores for each criterion

<BEGIN ANALYSIS>
(1) Personalized rubric with 1–5 scores for each criterion

Table 19: The prompt for $\mathcal{E}_p$

---

**Prompt for $\mathcal{E}_p$**

---

You are evaluating responses exactly like the specific human who wrote the examples. Replicate their preferences, strictness/leniency, tone, and feedback style precisely, including average score levels and feedback length from the examples.

First, recall the gold information need as the user's underlying intent, and use it as the primary reference for all judgments. Use the rubric below for scoring each criterion from 1 to 5 (Only integers are allowed). For each criterion, think step-by-step: (1) Identify key elements from the response, (2) Compare to examples and gold information need, (3) Assign score based on rubric, (4) Provide concise feedback mirroring example style.

Personalized rubric for this user and query:
{gold rubric}
Instructions:
- Personalize judgments to match the examples exactly; if patterns show leniency or strictness on any criterion (e.g., tone or interest alignment), apply similarly across all evaluations while referencing average scores from examples.
- Be concise and actionable in feedback. Mirror the example evaluator's language, politeness level, and any emojis precisely.
- First, think step-by-step for each criterion between <think> and </think> tags, write your thoughts. Then, provide the score and feedback.

<USER INPUT>
Query: {query}
Gold Information Need: {gold information need}
<END USER INPUT>

<EXAMPLES>
Response: {response in R-J pair}
Needs Alignment Score: {needs alignment score in R-J pair}
Needs Alignment Feedback: {needs alignment feedback in R-J pair}
Content Depth Score: {content depth score in R-J pair}
Content Depth Feedback: {content depth feedback in R-J pair}
Tone Score: {tone score in R-J pair}
Tone Feedback: {tone feedback in R-J pair}
Explanation Style Score: {explanation style score in R-J pair}
Explanation Style Feedback: {explanation style feedback in R-J pair}

...

<END EXAMPLES>

<EVALUATE USER INPUT>
Response: {new response}

<THINK>

---

Table 20: The prompt for feedback meta-evaluation

---

**Prompt for feedback meta-evaluation**

---

You are evaluating whether the Generated Feedback evaluates the response's {criterion} the way the Human Feedback author would.

Judge human-likeness by checking if the Generated Feedback:
- Generally reflects the Human Feedback's perspective: main priorities, focus areas, and overall severity/leniency
- Preserves the key issues and praise emphasized by the human; minor differences or additional nuance are acceptable if they do not distort the human's intent
- Is mostly accurate and grounded in the Query, Gold Information Need, and Response
- If the Generated Feedback reasonably emulates the human evaluator overall, even with minor differences, respond with "O".

Context
Query: {query}
Gold Information Need: {gold information need}
Response: {response text}
Rubric: {gold rubric}

Human Feedback (reference): {human feedback}
Generated Feedback (to judge): {generated feedback}

Decision
- If the Generated Feedback reasonably emulates the human evaluator overall, preserving the main intent and tone even with minor differences, respond with "O".
"X".

Respond with only one word: "O" or "X".

---

Table 21: The prompt for generating general profile

---

**Prompt for generating general profile**

---

You are to produce a user profile that helps with information-seeking tasks.
Information-seeking tasks aim to address a user's information need by providing the desired information in an appropriate form. Analyze the user's full search and chat history and infer preferences which help satisfy future personalized information seeking task for the user.
Based on the user's preferences, describe the user's profile in as much detail as possible that would help satisfy future information seeking tasks.
Express needs and preferences without relying on demographic labels (e.g., nationality, ethnicity, region); ground all claims in observable behaviors and content/format preferences.
Structure the profile to maximize usefulness for information seeking:

<OUTPUT GUIDELINES>
- Specify overall interests (broad and specific)
- For each interest:
1) frequently explored subtopics and focal aspects (from questions and follow-ups);
2) preferred presentation signals (organization, content depth, medium, tone, explanaition style) inferred from behavior;
3) personalization takeaways and actionable guidance (Treat language usage as behavior (not identity); avoid implying nationality or region)

---

Table 22: The prompt for selecting relevant history

| Prompt for selecting relevant history |
|---|
| You are a strict reviewer selecting history sessions that help personalize the response to THIS query. |
| Given the user's query and ONE history session (a chat file or a single search line), decide if this session contains concrete, query-specific personalization signals. |
| Explicit rule: If the session is related to the query and will help personalize the response to that query, you MUST accept it (set keep=true). If not, reject. |
| Keep ONLY if the session includes user-specific signals that would concretely change what the assistant says for this exact query: stable preferences (likes/dislikes), explicit constraints (budget/time/access), background (role/skill/industry), environment/config, or prior choices relevant to this topic. |
| If the history session is a past web search, it contains ONLY the search query string; judge relevance based on that query text alone (ignore any page content). |
| Important: Accept indirectly related but decision changing constraints as adjacent constraints. For example, for a game recommendation query, searches like 'GeForce Now', 'cloud gaming', 'Mac gaming', 'Steam Deck', 'low-end laptop', 'controller support', or 'RTX 3050 vs 3060' indicate platform/performance/input constraints that filter or change recommended titles. Classify such items as adjacent constraint and keep them. |
| Reject off-topic content or history unrelated to this query topic. |
| <Query>: {query} |
| <History Session>: {session text} |

Table 23: The prompt for generating query-aware profile

| Prompt for generating query-aware profile |
|---|
| You are a personalization profiler. |
| Goal: Create per-query personalized guidance on each criterion. |
| <QUERY> |
| <SEARCH HISTORY> |
| <CHAT HISTORY> |
| |
| <INSTRUCTIONS> |
| Do NOT default to general preferences. Extract query-topic-specific signals from the user's history: |
| |
| Repeated subtopics and facets specific to THIS query's topic (subcategories, approaches, features, constraints). |
| Format/style cues: preferred structure (checklists/templates/stepwise), concision vs detail, examples, references. |
| Negative/avoid cues: content types they ignore/dislike, off-topic tangents, categories or formats they avoid. |
| Determine domain fit: professional/work vs hobby/personal, using cues in the query and history. |
| |
| If professional/work: prioritize history mentioning projects, workplace tools/stacks, teams/clients, deadlines, compliance; de-prioritize hobby-only signals. |
| If hobby/personal: prioritize history about personal projects, de-prioritize enterprise/production requirements. |
| If mixed or ambiguous: assess the query's complexity/difficulty level to determine domain - complex/technical queries suggest professional context, simpler and easier queries suggest hobby/personal context. |
| Selectively cite only history aligned with the inferred domain; ignore high-frequency but off-domain signals. |
| |
| <OUTPUT GUIDELINES> |
| For EACH criterion (Need Alignment, Content Depth, Tone, Explanation Style), generate a comprehensive, highly detailed profile tailored to THIS specific query using only the most relevant history signals. |
| |
| Provide extensive detail on what to emphasize: include specific topics, subtopics, approaches, methodologies, tools, frameworks, and examples that align with the user's demonstrated interests. |
| Elaborate thoroughly on what to avoid: specify particular content types, tangents, approaches, terminology levels, or presentation styles that the user has shown disinterest in or actively filtered out. |
| Detail exactly how to deliver content: specify preferred formatting (bullet points, numbered lists, code blocks, tables), structural organization (step-by-step vs. overview-first vs. problem-solution), level of technical depth, use of examples, and pacing of information. |
| Specify the reasoning behind each profile choice: explain why certain approaches work better for this user based on their demonstrated patterns and preferences. |

Table 24: The prompt for selecting oracle history

---

**Prompt for selecting oracle history**

---

You are a strict reviewer selecting history sessions that help personalize the response to THIS query. Given the user's query and ONE history session (a chat file or a single search line), decide if this session contains concrete, query-specific personalization signals.

If a GOLD RUBRIC is provided, treat it as authoritative ground truth for this query's personalization targets. Favor sessions that directly instantiate, confirm, or constrain the rubric.
Explicit rule: If the session is related to the query and will help personalize the response to that query, you MUST accept it (set keep=true). If not, reject.
Keep ONLY if the session includes user-specific signals that would concretely change what the assistant says for this exact query: stable preferences (likes/dislikes), explicit constraints (budget/time/access), background (role/skill/industry), environment/config, or prior choices relevant to this topic.

If the history session is a past web search, it contains ONLY the search query string; judge relevance based on that query text alone (ignore any page content).
Important: Accept indirectly related but decision-changing constraints as adjacent constraints. For example, for a game recommendation query, searches like 'GeForce Now', 'cloud gaming', 'Mac gaming', 'Steam Deck', 'low-end laptop', 'controller support', or 'RTX 3050 vs 3060' indicate platform/performance/input constraints that filter or change recommended titles. Classify such items as adjacent constraint and keep them.
Reject off-topic content or history unrelated to this query topic.

Query: {query}
Gold Rubric: {gold rubric}
History Session:{history session}
...

---

Table 25: The prompt for CoT and Pseudo-history expansion in history retrieval

---

**Prompt for CoT expansion**

---

You are a retrieval assistant for user history (chat + search).
Given the original query, think step by step to infer the key aspects that seem important for retrieving relevant past sessions.

Query: {query}

---

No bullets, no extra text.

**Prompt for Pseudo-history expansion**

---

You are a retrieval assistant for user history (chat + search).
You are generating a piece of the user's past history entry (it can be either a chat message to an assistant or a search query with notes).
The generated text should sound like something the user actually wrote before, and it must be highly useful for retrieving relevant history sessions.

*Guidelines:*
– Write in natural style as if it were truly authored by the user.
– It may look like a chat utterance (full sentence, conversational) or a search query (short and keyword-based).
– Output ONLY the history text.

Query: {query}

---

History:

---

