# OpenReview forum: "BESPOKE: Benchmark for Search-Augmented Large Language Model Personalization via Diagnostic Feedback"
_ICLR.cc/2026/Conference — Submitted to ICLR 2026_

### Official Review · Reviewer_eGSd · 2025-10-27

**Soundness:** 3
**Presentation:** 3
**Contribution:** 3
**Rating:** 8
**Confidence:** 4

**Summary:**

The authors propose a new framework for evaluating search-augmented LLMs with a specific focus on user personalization in information-seeking tasks.

Specifically,
1. They collect interaction data between real human annotators and search systems (both search engine and search-augmented LLM).
2. Develop an annotation rubric and ask the same annotators to score and provide feedback on $k$ responses.
3. Build an evaluation pipeline for scoring (1) factuality and (2) personalization of a response conditioned on the user's past judgements (collected in the previous step).
4. Validate and ablate the evaluation pipeline against human-annotated labels.
5. Evaluate search-augmented LLMs using the proposed framework and analyze different context construction configurations.

Overall, the paper addresses the gap between evaluation of general search-augmented LLM evaluation and personalization.

**Strengths:**

The paper raises the importance of personalization when interacting with search-augmented LLMs for information seeking tasks.

- The paper discusses relevant prior work and clearly identifies a clear research gap (Section 2).
- The paper bases its evaluation framework on the data collected from natural unconstrained human user interactions with search systems. This approach is strongly preferred over synthetic and/or task-contsrained data collection (generation) settings (Section 3).
- The paper thoroughly validates and ablates its evaluation pipeline against human-annotated labels (Section 4.1).
- The paper states the importance and utilizes natural language feedback as an important signal for learning and evaluation in addition to numerical scores (Sections 3.3 and 3.4).
- The paper thoroughly studies and ablates different user context construction configurations (Section 4).
- The paper has a logical easy-to-follow narration flow.

**Weaknesses:**

### Factuality
- The proposed factuality metric is the recall over the atomic facts included in the **personalized** response, $r^+$. However, it should be possible to construct a highly relevant and factual response, $\hat{r}$, that would have minimal overlap with $r^+$. As factuality of search-augmented LLMs is not the main focus of the paper, I believe it would be best to either (1) rephrase or (2) remove factuality assessment from the pipeline.
- Additionally, general factuality is *hard* to define. If a fact can be found through a web search, does it make it factual? The current phrasing of the section can raise such questions, which does not directly relate to the main contributions of the paper.

### Impact of Web Search
In Section 4.5, the authors ablate the impact of the web search module (engine) on the system's performance, by comparing the base system with built-in web search tool and an LLM with access to *ground truth* documents as part of its context. The authors report improvements when the LLM is provided with less noisy higher quality context, with more pronounced improvements among reasoning models. The authors conclude that "reasoning models are more capable of analyzing and integrating the provided information". However, there are no significant differences between non-reasoning and reasoning models when provided with noisy context from a search engine (in some cases non-reasoning models score higher). Thus, a further analysis is required to understand this discrepancy, as the current conclusion is not supported by one of the settings.

**Questions:**

#### Suggestions for Improving the Flow and Clarity
- In Section 4.1 (Line 287), the authors can elaborate on w/o Feedback baseline to improve clarity.
- In Section 4.2 (Lines 317-322), make the distinction between (2) and (3) more clear.
- In Section 4.2, the authors directly discuss the results after presenting the four aspects of user context construction. Even though the details are included in the Appendix, a brief overview of the generation pipeline (e.g., how is user profile constructed) would be helpful in the main section of the paper.

### Learning from Feedback

The current evaluation framework relies on user annotations in the form of numerical scores and natural language diagnostic feedback. However, in real-world settings, we only have access to raw context in the form of search histories and chat interactions, and additional human-annotated labels are missings. A natural and interesting question arises: how can we build an evaluation framework that only has access to the raw context (no scores or diagnostic feedback)? The raw context can contain rich (but also noisy) feedback signals, which can be used for building user profiles used for both personalized evaluation and generation pipelines. Additional preliminary analysis would significantly strengthen the paper (but can also be thoroghly studied in a separate future work).

---

> ### Author Response · Authors · 2025-11-16
> **Initial response (1/2)**
>
> Thank you for your interest in our work and for the constructive suggestions that help improve it. We would like to provide responses to the points you raised. After you review our answers, if you feel that your concerns have been fully addressed, we will immediately upload an updated PDF with the proposed revisions.
>
> ## About Suggestions for Improving the Flow and Clarity
>
> - In the current version, we describe the w/o Feedback baseline in Appendix C.2. However, as you pointed out, the main text does not guide readers to this section, which may reduce clarity. To address this, we have added an explicit pointer to this baseline in Section 4.1, as you suggested.
> - Regarding the points related to Section 4.2, we kindly ask you to review the additional explanations we provide below. If these clarifications address your concerns, we will incorporate your suggestions into the revised PDF together with the updates mentioned above. Thank you again for offering thoughtful suggestions that help strengthen our work.
>
> ## About “factuality”
>
> Our intention was to use recall as an auxiliary metric that complements Need Alignment and Content Depth. Specifically, the role of recall was limited to measuring how well the response includes the key facts explicitly stated in the gold information. As you noted, the current term “factuality” goes beyond this intention and may cause confusion. **We fully agree with this point. To avoid misunderstanding, we will revise the metric name from “factuality coverage” to “gold information coverage,” which more accurately reflects its purpose.**

---

> ### Author Response · Authors · 2025-11-16
> **Initial response (2/2)**
>
> ## Impact of Search Quality on Reasoning vs. Non-Reasoning Models
>
> In Section 4.5, our goal was to convey two points. First, the quality of web search is highly important. Second, when search quality is high, reasoning ability becomes more influential. We did not intend to claim that reasoning models are universally superior at analyzing or integrating retrieved information in every scenario. **We acknowledge that our writing may have given the wrong impression, and we will revise this part to clarify our intended message.**
>
> Additionally, your comment inspired an interesting research question. Specifically, we became curious about **how the performance gap between reasoning and non reasoning models changes depending on search quality.** In Section 4.5, we assumed an ideal situation in which retrieval works well. Building on this, we conducted an additional experiment in which we gradually injected noise into the gold information to simulate lower quality search results. We then compared the models using the two metrics most sensitive to search quality, Need Alignment and Content Depth.
>
> (To introduce noise, we replaced a portion of the gold information for each user query with gold information taken from different users and different queries, according to a specified noise ratio. This allowed us to simulate increasingly noisy web searches.)
>
> Below is the result table from this additional experiment.
>
> | Model | Noise ratio | Need alignment | Content Depth | Avg | $ \Delta $ Avg |
> | :---- | :---- | ----- | ----- | ----- | ----- |
> | Sonar | 0 | 58.93 | 60.80 | 59.87 | \- |
> |  | 30 | 58.80 | 59.87 | 59.33 | \-0.90% |
> |  | 50 | 57.73 | 57.20 | 57.47 | \-4.18% |
> |  | 70 | 51.60 | 50.93 | 51.27 | \-16.78% |
> | Sonar-Reasoning | 0 | **60.13** | **62.67** | **61.40** | \- |
> |  | 30 | 60.13 | 60.13 | 60.13 | \-2.11% |
> |  | 50 | 54.93 | 55.20 | 55.07 | \-11.50% |
> |  | 70 | 49.47 | 50.27 | 49.87 | \-23.13% |
>
> **First, when only clean gold information is provided, the reasoning model shows highest performance. However, as noise increases, the reasoning model exhibits a sharper performance drop compared with the non-reasoning model, and its absolute performance eventually falls below that of the non-reasoning model.** This indicates that the reasoning model is **more sensitive to the quality of the context produced by web search.**
>
> Based on this finding, we plan to add a clarification that reasoning models are more dependent on the accuracy of web search results. **Reasoning models benefit when web search is accurate, but when web search contains noise, they can become disadvantaged relative to non reasoning models.** Therefore, achieving optimal personalization requires both reliable web search and advanced reasoning abilities.
>
> ## About “learning from feedback”
>
> We agree that the point you raised is important for advancing personalization research.
>
> We believe our human annotation based evaluator can serve high quality reference or reward model for training evaluators that rely only on raw context. To be specific, one possible approach is to treat the human annotation based evaluator as a teacher. The evaluator that receives only weak signals from raw search and chat history can then be trained to mimic the teacher’s scoring function or policy using methods such as distillation or reward model based reinforcement learning.
>
> However, we also believe that this line of research is broad enough to constitute a standalone direction. For this reason, we consider it beyond the scope of the current paper and leave it as an important area for future work. **We would like to emphasize that, when pursuing this direction, our benchmark and our human annotation based evaluator can serve as a high quality reference of personalized evaluator.**

---

### Official Review · Reviewer_RMtA · 2025-10-29

**Soundness:** 3
**Presentation:** 3
**Contribution:** 2
**Rating:** 4
**Confidence:** 4

**Summary:**

This paper addresses an important but underexplored question: how to evaluate *personalization* in search-augmented LLMs.

The authors propose **BESPOKE**, a human-curated benchmark that captures authentic chat and search histories from real users, pairs queries with gold information needs, and collects multi-criteria feedback (need alignment, content depth, tone, and explanation style).

BESPOKE provides both *factual* and *personalization* evaluations, offering diagnostic feedback for analyzing where models succeed or fail. Experiments on major search-augmented LLMs show consistent gains from user-context conditioning and highlight personalization as an ongoing challenge .

**Strengths:**

- **Timely and relevant topic:** Personalization in RAG systems is highly relevant as LLMs shift from generic to user-aware reasoning.
- **Annotation effort:** The benchmark construction is carefully designed and grounded in long-term, real human histories, ensuring realism.

**Weaknesses:**

1. **Lack of conceptual clarity on personalization.**

   The central concept of *personalization* is never explicitly or rigorously defined in the paper. It remains unclear whether personalization refers to modeling *user persona* (i.e., stable, identity-dependent traits and preferences), *long-context reasoning* over past dialogue turns, or simply *conditioning* on user-specific interaction history. Without a theoretical or formal framework delineating these possibilities, it is difficult to interpret what kind of personalization BESPOKE is truly evaluating. From my understanding, in human–computer interaction, personalization involves sparse, implicit, and often latent signals that go beyond explicit interaction history. Treating user history as equivalent to persona modeling risks conflating two fundamentally different constructs—short-term contextual adaptation versus long-term user modeling. A clearer conceptual grounding is therefore essential.

2. **Ambiguity between personalization and multi-turn dependency.**

   Many of the benchmark’s manipulations—such as whether to provide user history, whether to use full or summarized context, and whether to inject selective history—could equally be interpreted as probing *multi-turn dialogue reasoning* rather than personalization. Multi-turn dialogue modeling inherently requires tracking semantic dependencies across turns, which can produce similar quantitative effects as “personalized” context conditioning. To convincingly claim that BESPOKE measures personalization, the authors must clarify how their benchmark distinguishes between these two phenomena. For instance, what aspects of the benchmark ensure that performance gains stem from user-specific adaptation rather than generic long-context coherence?

3. **Personalization–factuality tension in RAG.**

   The paper situates personalization within a RAG framework, which typically targets factual or knowledge-seeking tasks. However, this raises a conceptual tension: factuality-based information retrieval emphasizes *objective correctness* and *information need satisfaction*, whereas personalization emphasizes *subjective preference alignment*. In tasks like “What are the top-10 cited CS papers in 2025?”, all users should receive the same answer; personalization is not only unnecessary but potentially misleading. The paper should clarify under what task types or conditions personalization is meaningful in RAG, and whether factual information retrieval is even the right domain for testing personalized behavior. A deeper justification of why personalization matters in factual contexts is needed.

4. **Feedback–persona mismatch and empirical bias.**

   In Stage 2 of benchmark construction, user feedback is iteratively incorporated to refine LLM responses into “gold” answers. However, these feedback signals appear to be task-specific—focused on improving the answer for a given question—rather than generalizable reflections of user persona or preference. Using such localized, task-specific feedback to evaluate general personalization ability risks introducing empirical bias: models may learn to fit task-level optimization signals rather than genuine personalization behaviors. The authors should discuss this potential mismatch and clarify whether the benchmark measures general persona alignment or task-specific adaptation.

5. **Metric scope and universality.**

   The proposed four evaluation dimensions—*need alignment*, *content depth*, *tone*, and *style*—may not be equally applicable across all tasks. For many short factual queries, such as “How many countries are in Europe?”, dimensions like tone and style are largely irrelevant and may confound evaluation. This raises questions about the universality and discriminative power of these metrics: are they intended to cover all information-seeking interactions, or only subjective or conversational ones? The authors should clarify whether these dimensions are weighted or selectively applied depending on task type, and justify their inclusion as general indicators of personalization.

**Questions:**

1. Could you clearly define *personalization* in your framework? Is it about modeling user persona, context dependency, or preference learning?
2. How does BESPOKE ensure that it measures personalization rather than generic multi-turn contextual reasoning?
3. What is the motivation for studying personalization in RAG scenarios where factual accuracy dominates?
4. How does the feedback process (iterative user-LLM refinement) avoid conflating task-specific optimization with genuine persona alignment?
5. Are all four evaluation criteria equally relevant across information-seeking tasks? Could some be task-specific?

---

> ### Author Response · Authors · 2025-11-16
> **Initial response (1/2)**
>
> Thank you for your interest in our work. We would like to provide responses to the points you raised. It is possible that we may have misunderstood part of your comment, and if additional clarification is needed, we are more than happy to provide further answers.
>
> ## About personalization in our work (W1, W2, Q1, Q2)
>
> Our benchmark focuses on personalization in information seeking tasks. In our setting, personalization refers to **the ability to generate responses that address a user’s information needs while reflecting the user’s search and chat history, preferences, and favored styles of composition, explanation, and presentation.** This definition aligns with prior works \[1\]\[2\]\[3\], and our benchmark is designed to evaluate this capability.
>
> We think concepts you mentioned, such as modeling user persona, context dependency, preference learning, and multi turn contextual reasoning, **as methodological choices for implementing personalization**. These techniques may help a system perform better on the type of personalization that our benchmark targets, but **our benchmark itself does not belong exclusively to any one of these categories.**
>
> To evaluate such personalization, our evaluation framework uses user specific and query specific evaluation criteria, represented by the R-J pairs and gold rubrics described in Section 3.4. This makes our benchmark fundamentally different from traditional benchmarks that evaluate generic behavior, and enables effective assessment of personalized information seeking.
>
> We also do not assume that user history and user persona are identical. As you noted, personas often contain sparse and latent characteristics that extend beyond observable interaction logs. In our benchmark, **user histories serve as observable proxies that allow models to infer such latent preferences**. The model must interpret the “**implicit preference”** expressed in the search and chat history, infer the preferences of the user, and produce responses accordingly.
>
> ## About our motivation (W3, Q3)
>
> The motivation behind our work is **that satisfying a user’s information needs requires personalization that reflects the user’s background, interests, and preferred modes of expression.** Even for the same query, the information that should be provided differs depending on the user’s context and interests \[4\]. For instance, in Figure 1, both users ask the same question about the difference between EVs and hybrids. However, User 1 has shown persistent interest in environmental and carbon emission topics, so environmental and sustainability aspects become key information. User 2 has repeatedly shown interest in vehicle performance and driving characteristics, so acceleration, driving feel, and powertrain become the central information. If such background and interests are ignored, the system may provide irrelevant or less important information, ultimately failing to address the user’s information need.

---

> ### Author Response · Authors · 2025-11-16
> **Initial response (2/2)**
>
> ## About our tasks and metrics (W5, Q5)
>
> We fully agree that short factual queries leave no room for personalization, since the correct answer does not change across users. **Therefore, our benchmark intentionally excludes short factual queries. It focuses instead on information seeking tasks where personalization meaningfully affects the quality of the response, specifically analysis, guidance, recommendation, and explanation.** These categories were defined based on the descriptions provided to annotators during query creation, and are adapted from the description in \[5\].
>
> - Analysis: Requests analysis of a topic, including comparison, evaluation, consideration of multiple perspectives, and synthesis of information from several sources.
> - Guidance: Requests procedures, instructions, or practical advice for completing a task, often involving step by step reasoning or problem solving.
> - Recommendation: Requests suggestions tailored to specific constraints, preferences, or criteria.
> - Explanation: Requests detailed conceptual or educational explanations aimed at deeper understanding.
>
> Queries in these categories have answers that legitimately vary depending on the user. For instance, the example you mentioned, “How many countries are in Europe”, yields the same answer regardless of who asks. In contrast, analysis queries like “the difference between EVs and hybrids” vary with user background, as illustrated in Figure 1\. **Annotators created queries only within the four task categories above, and after a multi stage review, we excluded any short factual queries from the benchmark.**
>
> Under this setting, our four evaluation criteria need alignment, content depth, tone, and explanation style are not intended as universal metrics for all forms of information seeking. **Instead, they are designed to evaluate different aspects of personalization in tasks where personalization is genuinely meaningful**. Since the tasks in BESPOKE all belong to these categories, we believe that these four criteria are valid and informative for evaluating personalization in information seeking.
>
> ## About constructing gold responses (W4, Q4)
>
> The iterative feedback process in Stage 2 may appear to be a task specific procedure focused on improving individual answers, but our goal is broader. **We aim to capture not only general preferences but also topic conditioned preferences that manifest only in certain types of queries.**
>
> We believe that an i**ndividual’s preferences are better understood as a set of conditional preferences that vary across topic and context.** For example, a user may prefer concise yet technically deep explanations for topics related to their area of expertise, but prefer a more casual and friendly presentation for hobby related or lightweight topics \[6\]\[7\]. We think the collection of these topic dependent preference patterns as the full expression of a user’s overall preference profile.
>
> Furthermore, the iterative refinement in Stage 2 is not merely about improving answer quality. **It is a process through which annotators reflect their own preferences while constructing the gold responses.** **We instructed annotators explicitly to judge based on their own information needs and personal preferences, rather than to search for a generally good or generic answer.** This ensures that the resulting gold response and rubric faithfully reflect the preferences of the actual user.
>
> \[1\] Lamp-qa: A benchmark for personalized long-form question answering; Salemi et al.; EMNLP 2025
> \[2\] LongLaMP: A Benchmark for Personalized Long-form Text Generation; Kumar et al.; arXiv 2025;
> \[3\] Lamp: When large language models meet personalization; Salemi et al.; ACL 2023
> \[4\] To personalize or not to personalize: Modeling queries with variation in user intent; Teevan et al; SIGIR 2008
> \[5\] Search Arena: Analyzing Search-Augmented LLMs; Miroyan et al; arXiv 2025
> \[6\] Characterizing the influence of domain expertise on Web search behavior; White et al; WSDM 2009
> \[7\] Characterizing Web Content, User Interests, and Search Behavior by Reading Level and Topic; Kim et al; WSDM 2012

---

### Official Review · Reviewer_efh6 · 2025-10-31

**Soundness:** 4
**Presentation:** 3
**Contribution:** 3
**Rating:** 2
**Confidence:** 4

**Summary:**

This paper introduces a new benchmark called BESPOKE, designed to evaluate how well search-augmented large language models (LLMs) can personalize information-seeking responses. Specifically, it is constructed with 30 annotators alongside their search and chat histories over three weeks, and further paired with 150 user-authored queries that include gold information needs and fine-grained feedback. Then, the experiments on multiple search-augmented LLMs show that leveraging user history (especially query-aware, profile-based contexts) enhances personalization, with challenges on inferring user needs (from long-term histories) and improving reasoning over retrieved information remaining.

**Strengths:**

* The proposed benchmark on personalization in search-augmented LLMs is timely, which should be highly valuable to the community. Personally, I feel this area has been underexplored despite its growing practical importance.
* BESPOKE is constructed with both the search and chat histories of multiple users for three weeks, which is realistic and reflects relatively long-term interactions over existing benchmarks.
* The experiments are comprehensive, including multiple state-of-the-art models and several useful analyses on where the performance improvement comes from.

**Weaknesses:**

* The collected 2870 user history sessions (combining both the search and chat) over 30 users for 3 weeks seem very small (which corresponds to 4-5 instances per user for one day), from which it is questionable whether the interactions occur and are collected in sufficiently realistic settings, as typical search and chat users make much more queries per day.
* While the authors claim that the annotators have diverse backgrounds, no concrete justification or demographic evidence is provided.
* The benchmark focuses primarily on proprietary systems; however, they are typically not very reproducible. I would like to suggest including open-weight models (or research-grade models), to further facilitate the reproducibility of this benchmark.
* The results in the main table show similar performances across different models, which deserve deeper discussions. One speculation is that, for this personalization benchmark, the capability of models may matter less compared to the contextual information provided.
* As a benchmark study, it would be beneficial to more explicitly list out the limitations of existing models and potential future works to mitigate them.

I am happy to increase the score to accept, if the above issues and the ethical concern (below) are fully addressed.

**Questions:**

Please see Weaknesses above.

**Details Of Ethics Concerns:**

The proposed benchmark collects search and chat histories of users; thus, the re-identification risk might be very high (cf. the well-known AOL search-log incident). I think the main reason that many institutions have not released the datasets for the search and chat histories of users is the difficulty of ensuring true anonymization and privacy protection: even after removing direct identifiers, seemingly harmless query or conversation patterns can still reveal personal information through linkage attacks. Therefore, more deeper investigation into it is needed before published.

---

> ### Author Response · Authors · 2025-11-16
> **Initial response (1/2)**
>
> Thank you for your thoughtful and constructive review. We hope our responses fully address your concerns. If there are any further questions or clarifications needed, we are more than willing to address them.
>
> ## BESPOKE has sufficient interaction histories (W1)
>
> For chat histories, some users start a new session whenever the topic changes, but many others continue within the same session even when shifting to entirely different topics. Several of our annotators also exhibited this information-seeking style. In such cases, multiple topics become mixed within a single session, often resulting in chat sessions exceeding 25 turns. Because these heterogeneous session-usage patterns exist, simply counting the number of sessions is not an appropriate way to assess data scale. **In fact, when we aggregated all chat turns, we observed a total of 2,983 turns. Considering this alongside the 2,153 search sessions in the search history, we believe that the amount of collected data is sufficiently large.**
>
> ## Diverse background of BESPOKE annotators (W2)
>
> We define diverse backgrounds as **a variety of professional expertise and topical interests that directly influence annotators’ information-seeking behavior.** To ensure such diversity, we conducted a screening process in which applicants reported their expertise area (the domain related to their current occupation) and their interest area (hobbies or other personal interests). Using this information, we selected annotators such that neither their expertise nor their interest areas overlapped significantly with previously recruited annotators. This allowed us to maximize diversity within the annotator pool while maintaining representativeness. For privacy protection, we collected only domain-level expertise and interest categories, and did not request any sensitive personal attributes such as age or gender.
>
> Below is the full set of expertise and interest areas represented in our annotator pool.
>
> **Expertise area**: Linguistics, Pharmaceutical Science, Law, Cybersecurity, AI, Sales Management, Music A\&R, Statistics, Medical, Sports Industry, Hardware, Dental, HCI, Physics, Entertainment, Biology, Public Administration, Electronics Engineering, F\&B, Industrial Engineering, Materials Engineering, Business Analysis, Marketing
>
> **Interest area**: health, games, economy, performing arts, engineering, science, education, finance, technology, academic papers, news, data science, reading, videos, design, marketing, restaurants/food, document writing, literature, art, delivery, translation, crime, law, security, volunteering, real estate, beauty, business, society, social science, industry, software, shopping, sports, animation, travel, history, foreign language learning, movies, entertainment, cooking, fortune-telling, music, medicine, artificial intelligence, humanities, interior design, automobiles, electronics, politics, religion, alcoholic beverages, startup, employment, computer hardware, computer science, statistics, telecommunications, trends, fashion, programming, academia, academic research, environment
>
> The interest areas correspond to the normalized high-level categories defined in Appendix B.1. To assess whether annotators’ interests were concentrated in only a few topics, we computed Shannon’s equitability over the frequency distribution of interest areas. The resulting value of 0.91, close to the upper bound of 1, indicates that the distribution is highly even. In other words, no small subset of topics dominates the pool; instead, interests are broadly and uniformly represented. This demonstrates that, along the dimensions of topical interests that influence information-seeking behavior, our annotator pool exhibits substantial diversity.
>
> ## Open-weight models can also serve as effective evaluators. (W3)
>
> We additionally evaluated whether our evaluator could be replaced by open-weight models (GPT-oss 120B and Qwen3-235B-A22B-Thinking-2507). The results are presented below.
>
> | Evaluator | Criterion | Pearson | Spearman | Feedback Acc. |
> | :---- | :---- | ----- | ----- | ----- |
> | GPT-5 | Need Align. | 0.881 | 0.882 | 0.85 |
> |  | Content Depth | 0.845 | 0.855 | 0.87 |
> |  | Tone | 0.9 | 0.893 | 0.903 |
> |  | Style | 0.784 | 0.797 | 0.857 |
> |  | Avg. | 0.853 | 0.857 | 0.870 |
> | GPT-OSS 120B | Need Align. | 0.868 | 0.868 | 0.82 |
> |  | Content Depth | 0.827 | 0.831 | 0.787 |
> |  | Tone | 0.852 | 0.839 | 0.847 |
> |  | Style | 0.72 | 0.742 | 0.803 |
> |  | Avg. | 0.817 | 0.82 | 0.814 |
> | Qwen3-235B-A22B-Thinking-2507 | Need Align. | 0.879 | 0.883 | 0.81 |
> |  | Content Depth | 0.828 | 0.842 | 0.77 |
> |  | Tone | 0.866 | 0.86 | 0.843 |
> |  | Style | 0.769 | 0.786 | 0.743 |
> |  | Avg. | 0.836 | 0.843 | 0.792 |
>
> **These results demonstrate that open-weight models can also serve as effective evaluators.** We will include this finding in the revision by adding a discussion of how such open-weight models can be used as alternative evaluators within our evaluation pipeline.

---

> ### Author Response · Authors · 2025-11-16
> **Initial response (2/2)**
>
> ## Influence of Backbone Model Capabilities on Personalization Performance (W4, W5)
>
> First, the **backbone model naturally influences the overall performance range.** As shown in Table 4, stronger models such as **o3-search** tend to exhibit consistently higher performance across metrics, whereas relatively weaker models such as **GPT-4o-search** show a lower performance range. This suggests that the inherent capability of the model sets the upper and lower bounds of achievable personalization quality, and that the effectiveness with which user context is utilized determines how close the model can get to that bound.
>
> It is therefore important to discuss *which specific model capabilities* are responsible for determining this baseline performance range. Because all models we use are black-box systems, a detailed internal analysis is not possible. However, the most reasonable hypothesis is that both **reasoning capability** and **instruction-following ability** play critical roles.
>
> Interestingly, Gemini 2.5 Pro did not exhibit as strong a performance range as we initially expected. This stands in contrast to the performance of o3 on benchmarks such as *MMLU-Pro* and *Humanity’s Last Exam*, where the two models show comparable results. While investigating potential causes for this discrepancy, **we found consistent evidence that the Gemini family is substantially weaker than o3 on explicit instruction-following benchmarks such as IFEval and IFBench** (IFEval: 65.4 vs. 95.0; IFBench: 52.3 vs. 69.3).
>
> When using black-box models, all personalization contexts are provided to the model as **natural-language instructions**. The model should read these instructions and (1) infer the user’s information need, (2) retrieve appropriate evidence, and (3) generate a final response that reflects the user’s preferred focus, depth, and stylistic preferences. This entire process depends critically on how well the model can follow complex, multi-step constraints specified purely in text.
>
> **If a model exhibits weak instruction-following ability, it becomes more likely to misinterpret or underutilize user-specific cues embedded in the context, leading to degraded personalization throughout both the retrieval and response-generation stages.** We believe that this mechanism plausibly explains why backbone models differ so markedly in their achievable performance ranges.
>
> A more systematic investigation of this hypothesis is left for future work. We appreciate you for raising this insightful point\!
>
> ## About Ethics concerns
>
> We thoroughly remove all personally identifiable information from interaction histories is inherently challenging. We fully agree with the importance and difficulty of this task, and for that reason we applied a rigorous multi-stage de-identification process in which **each history was reviewed at least four times by different individuals**.
>
> To prevent any trace of identifiable information from remaining, we followed the two-stage procedure described in Appendix B.2. Importantly, when we determined that simply masking or modifying certain tokens was insufficient to eliminate potential re-identification risk, we removed the **entire session** from the dataset.
>
> This process involved several steps:
>
> 1. Annotators self-reviewed their own histories, **voluntarily removing any sessions they considered sensitive.**
> 2. Our team then manually inspected every session, admitting it into the dataset only **when all members unanimously agreed that no re-identification risk remained.**
> 3. In addition to manual inspection, **we applied LLM-based screening tools** to detect any potential PII, using multiple repeated passes to further minimize risk.
>
> Through this combination of human review, annotator self filtering, repeated LLM based inspection, and conservative session level removal, we ensured that all histories in the final dataset satisfied strict de-identification standards.

---

> ### Author Response · Authors · 2025-11-26
> **Additional response for ethics concerns**
>
> As an additional clarification regarding the ethical concerns raised, we first outline how we addressed information that could directly or indirectly identify individuals. During de-identification, we removed content that we judged to be potentially identifying, following the principles outlined in prior work such as WildChat \[1\]. Specifically, we excluded the following categories:
>
> - Personal names
> - Fine grained residential or location details (for example, exact addresses)
> - Personal identification numbers (for example, passport numbers or social security numbers)
> - Links to the personal social media account
> - Specific activities or events tied to a particular individual or time period
> - Any other sensitive personal information that could reveal or uniquely pinpoint an individual
>
> **To further account for the possibility that a combination of seemingly harmless user histories (even after removing direct identifiers) could still reveal a specific individual, we performed an LLM-based screening step**, which was mentioned in our initial response. For each user, we provided the entire history sessions to the model and asked whether the history presented any risk that a specific individual could be identified. **If LLM detected a potential risk of indirect re-identification through the combination of seemingly harmless histories or interaction patterns, we removed the entire set of associated sessions rather than masking individual tokens.** This screening and removal cycle was repeated until no identifiable patterns remained, ensuring that the retained histories carried no reasonable risk of linkage based identification.
>
> **We would also like to note that all data collection, review, and de identification procedures were formally reviewed, approved, and overseen by our Institutional Review Board (IRB). The IRB thoroughly evaluated our anonymization workflow**, including annotator self reviewing, multi stage human review, LLM-based screening, and conservative session level removal, and verified that each step met stringent privacy and human subject protection standards. **Through this IRB approval and oversight, we made every effort to minimize any potential risk of personal information being revealed in our dataset.** We are happy to address any further questions you may have.
>
> \[1\] WildChat: 1M ChatGPT Interaction Logs in the Wild; Zhao et al; ICLR 2024

---

### Official Review · Reviewer_AM4H · 2025-11-11

**Soundness:** 3
**Presentation:** 3
**Contribution:** 3
**Rating:** 6
**Confidence:** 4

**Summary:**

This paper presents BESPOKE, a benchmark designed to evaluate personalization in search-augmented LLMs. The dataset comprises 2870 real user search and chat sessions collected over a three-week period from 30 users, along with 150 user-authored queries accompanied by gold information needs and detailed user judgments (including both scalar ratings and free-form diagnostic feedback) over LLM-generated responses. Beyond providing a dataset, the paper introduces an evaluation framework built on GPT-5 that generates query-specific rubrics and uses an LLM judge to assess model outputs. Using this benchmark, the authors analyze the performance of several frontier search-augmented LLMs and extract insights into the current strengths, limitations, and bottlenecks of personalization in information-seeking settings.

**Strengths:**

- High-quality, realistic dataset. The benchmark offers rich, deeply contextualized real-world histories and multi-dimensional annotations (scores + diagnostic feedback), which are costly and difficult to collect at scale. This level of contextual grounding is rare and likely to be highly valued by the community.
- Provides both the raw data and a full end-to-end evaluation framework, enabling systematic and reproducible assessment of personalization in search-augmented LLMs.
- Insightful empirical findings. The analysis identifies concrete points of failure and success, e.g., the importance of query-aware context construction, the limits of current web-search quality, key reasoning bottlenecks, and the effectiveness of retrieval methods.
- Overall, the data collection, annotation protocol, experimental setup, and analyses appear thoughtful and well executed.

**Weaknesses:**

- Limited scale of queries. Although the dataset is rich, the benchmark includes only 150 test queries across 30 users, which is small compared to existing evaluation suites. This raises questions about coverage and diversity of personalization scenarios, although the depth of per-user context partly compensates for the small query set.
- Remaining self-preference concerns. While GPT-5 is used to reduce self-preference effects for Gemini-generated data, parts of the annotation pipeline still involve Gemini-generated responses, and GPT-family models ultimately evaluate GPT-family models (e.g., o3, GPT-4o). Additionally, the heavy reliance on GPT-5 across many stages (evaluation, rubric generation, gold information extraction, profile generation, history selection, and oracle annotation) may limit generalizability across model families and impose high computational cost for running the benchmark end-to-end.
- Underspecified evaluation procedure. Section 4.2 provides only a brief description of how model evaluation is operationalized. The absence of a clear full picture of the entire evaluation pipeline makes it difficult to independently verify or reason about the experimental setup and results. For instance, profile generation is described only by providing prompts in the appendix; there is no discussion of profile quality, consistency, or what kinds of user attributes or preferences are typically captured.
- Insufficient detail on data collection. The paper does not describe what instructions the annotators received for generating chat histories, search histories, or queries. Without this, it is difficult to infer the distributional characteristics of the dataset. Similarly, details about how annotation quality was validated (beyond basic self-consistency checks) are sparse.

**Questions:**

- Why is recall-only evaluation used for factual coverage in Section 4.1? With recall as the main metric for factuality, models that produce long responses may be rewarded despite adding inaccurate or irrelevant information. Is there a safeguard against this behavior? Precision seems relevant given that models often add extraneous information. What motivated the choice to exclude precision-oriented metrics?
- What value of k was used for generating hypothetical information needs? The text mentions sampling k responses but does not specify the actual hyperparameter.
- Did you analyze how the amount or type of user context affects personalization performance? For example, does increasing the number of sessions, the mix of search vs. chat history, or the recency of histories correlate with improvements?
- Can you provide a cost breakdown for running BESPOKE? This includes human annotation, LLM-assisted annotation, evaluation, profile generation, and search grounding. A per-query estimate would be particularly helpful for researchers considering extending the dataset.

**Details Of Ethics Concerns:**

In Appendix B.1, the paper states only that “all compensation adheres to ethical and legal standards, as verified through review by legal experts,” without disclosing the actual compensation amount or the structure. Given that annotators contributed three weeks of history collection and annotation, it is important to specify how they were compensated over this period. Providing more transparency seems needed.

The relevant excerpt from the appendix reads:
"Each recruited annotator receives a dedicated Google account created for research purposes, with a paid Google AI Pro plan. We also maintain ongoing communication channels between annotators and researchers to promptly address any questions arising during data collection or annotation processes. Annotators benefit from complimentary access to the AI Pro plan during the history collection period, along with adequate compensation for their annotation efforts. All compensation adheres to ethical and legal standards, as verified through review by legal experts. Furthermore, annotators receive comprehensive information on compensation and data collection procedures in advance, and we recruit only those who provide informed consent."

---

> ### Author Response · Authors · 2025-11-16
> **Initial response (1/3)**
>
> We sincerely appreciate your thoughtful and constructive feedback\!
>
> ## Dataset coverage and diversity (W1)
>
> We acknowledge your concern that the benchmark size is smaller than some existing benchmarks. However, substantial effort was devoted to ensuring diversity and representativeness of the user pool.
>
> To ensure diversity, we conducted a screening process in which we collected applicants’ **expertise area** (fields related to their current occupation) and **interest area** (hobbies and other personal interests). During the pre-screening stage, we collected both dimensions from the applicants and selected annotators such that neither their expertise nor their interest areas overlapped significantly with previously recruited annotators.
>
> As a result, the final annotator cohort spans **more than 22 professional domains** (including linguistics, pharmacy, law, cybersecurity, medicine, physics, industrial engineering, electrical engineering, marketing, etc.) and **over 60 topical interest areas** (health, gaming, economics, education, science, finance, literature, cuisine, crime/law, travel, history, arts, AI, statistics, hardware, fashion, environment, etc.). Thus, despite having only 30 annotators, the group is highly diverse across two independent dimensions. Please refer to the response for efh6, About User Diversity section for more details.
>
> In addition, to ensure that no small set of interests dominated the cohort, we also computed **Shannon’s equitability** over annotator interest distributions. As denoted in Section 3 and Appendix B.1., the equitability score was **0.91**, indicating that interest areas are evenly spread and not concentrated in a few topics.
>
> ## Open-weight models can also serve as effective evaluators. (W2)
>
> To verify whether our evaluator (GPT-5) can be replaced by open-weight models, we conducted experiments using two strong open-weight alternatives; GPT-OSS-120B and Qwen3-235B-A22B-Thinking-2507 as evaluators.
>
> | Evaluator | Criterion | Pearson | Spearman | Feedback Acc. |
> | :---- | :---- | ----- | ----- | ----- |
> | GPT-5 | Need Align. | 0.881 | 0.882 | 0.85 |
> |  | Content Depth | 0.845 | 0.855 | 0.87 |
> |  | Tone | 0.9 | 0.893 | 0.903 |
> |  | Style | 0.784 | 0.797 | 0.857 |
> |  | Avg. | 0.853 | 0.857 | 0.870 |
> | GPT-OSS 120B | Need Align. | 0.868 | 0.868 | 0.82 |
> |  | Content Depth | 0.827 | 0.831 | 0.787 |
> |  | Tone | 0.852 | 0.839 | 0.847 |
> |  | Style | 0.72 | 0.742 | 0.803 |
> |  | Avg. | 0.817 | 0.82 | 0.814 |
> | Qwen3-235B-A22B-Thinking-2507 | Need Align. | 0.879 | 0.883 | 0.81 |
> |  | Content Depth | 0.828 | 0.842 | 0.77 |
> |  | Tone | 0.866 | 0.86 | 0.843 |
> |  | Style | 0.769 | 0.786 | 0.743 |
> |  | Avg. | 0.836 | 0.843 | 0.792 |
>
> The results show that both models yield evaluations highly aligned with GPT-5, demonstrating that the benchmark **generalizes** across model families, and the **evaluation cost** can be substantially reduced by adopting open-weight evaluators.
>
> We will include this finding in the camera-ready version by adding a discussion of how such open-weight models can be used as alternative evaluators within our evaluation pipeline.
>
> ## About user profile (W3)
>
> Our experiments include two types of user profiles: **general profiles** and **query-aware profiles**.
>
> **General profile** summarizes a user’s entire search and chat history to capture broad, long-term preferences (e.g., topic interests, preferred information formats, explanation styles). While effective for modeling global tendencies, it may obscure fine-grained preferences. For example, if a user overwhelmingly queries work-related topics in a formal tone, their preferences for casual tone in hobby-related queries may be overshadowed.
>
> **Query-aware profile** extracts only the subset of user preferences relevant to the *current* query. For instance, for a work-related query, it emphasizes the user’s preference for formal explanations; for a hobby-related query, it surfaces their preference for a more casual tone.
> This allows models to adaptively apply personalization at the appropriate granularity.

---

> ### Author Response · Authors · 2025-11-16
> **Initial response (2/3)**
>
> ## Data collection details (W4)
>
> We detail here the instructions given to annotators during history, query, and judgment collection.
>
> ### **Instruction for history collection**
>
> Annotators were instructed to record search and chat histories **exactly as they normally would**, without prompting them toward specific topics or styles. Minimal intervention was used to preserve naturalistic behavior. To avoid unrealistic patterns, we provided examples of typical everyday information-seeking behaviors, ensuring the collected histories reflect real-world user signals suitable for personalization evaluation.
>
> ### **Instruction for query collection**
>
> Annotators were given four predefined information-seeking tasks (analysis, guidance, recommendation, explanation), along with multiple example queries for each task.
>  They were instructed to phrase queries naturally, just as they would when using ChatGPT or Gemini. Each query also required a **gold information need**, which annotators were asked to describe as explicitly and concretely as possible.
>
> ### **Instruction for judgment collection**
>
> For each query, annotators were given **six responses** (one gold response \+ five randomly sampled responses). For each response, annotators provided a scalar rating, and a free-form diagnostic explanation. Explanation of each rubric was shown for every annotation page to ensure consistent interpretation and reduce subjectivity.
>
> ### **Annotation quality control**
>
> We implemented multiple layers of quality assurance:
>
> * **Query stage**: Queries inconsistent with task definitions were returned for revision. Gold information needs were manually inspected and refined when insufficiently specific.
>
> * **Judgment stage**: Feedback that was too short or lacked concrete justification triggered revision requests.
>   This process ensured consistency, clarity, and high-quality annotations across the dataset.
>
> ## About factual evaluation (Q1)
>
> As you point out, using recall for factual coverage raises the concern that a model could obtain higher scores simply by generating longer responses that contain more content, including potentially inaccurate or irrelevant information. We agree that this is a valid risk, and we considered it carefully when designing our evaluation.
>
> Our motivation for using recall is that the purpose of factual coverage in Section 4.1 is to assess **whether the model includes all key facts** specified in the gold information need. In this setting, recall is the metric that most directly captures factual completeness.
>
> Although we considered precision-oriented metrics, we found that the combination of recall with our personalized metrics provides a natural safeguard against verbosity-driven inflation. Specifically:
>
> * **Need Alignment** penalizes responses that introduce content unrelated to the user’s information need.
>
> * **Content Depth** penalizes explanations that are shallow, unfocused, or excessively broad.
>
> Because long but unfocused responses score poorly on these dimensions, they do not yield inflated overall scores even if their factual recall is high. As a result, the evaluation remains robust while still emphasizing the primary goal of verifying factual completeness.
>
> ## Experimental detail (Q2, Q4)
>
> For generating hypothetical information needs, we used **k \= 6**. The cost breakdown for the full pipeline is as follows.
>
> Human annotation cost : $50 per annotator
>
> - $25 for history construction
> - $25 for query & judgment annotation
>
> LLM-assisted annotation cost (GPT-5 (reasoning effort \= high))
>
> - Gold rubric generation: **$0.015 per query**
> - Gold information need generation: **$0.005 per query**
>
> Total annotation cost per query: **\~ $0.35**
>
> Evaluation cost (our evaluator)
>
> - Personalization quality evaluation: **$0.05**
> - Factual coverage evaluation: **$0.05**
>   (based on the cost of evaluating 1 response against \~20 gold claims)

---

> ### Author Response · Authors · 2025-11-16
> **Initial response (3/3)**
>
> ## Additional analysis about user context (Q3)
>
> We analyzed whether having a larger number of user history sessions leads to greater improvements in personalized response quality. To do so, we computed the performance difference between the o3-search model’s w/o history setting and its w/ selected history profile setting on a per-user basis, and measured the correlation between this improvement and each user’s history session count. The Pearson and Spearman correlations were **0.35 and 0.26**, respectively, indicating that the number of sessions alone does not sufficiently explain gains in personalization performance.
>
> Next, to examine whether the type of history a user predominantly produces affects personalization, we calculated each user’s search-to-chat ratio (search session count / chat session count). The Pearson and Spearman correlations between this ratio and performance improvement were **0.33 and 0.19**, respectively. Thus, the dominant history type was also not a meaningful explanatory factor.
>
> Motivated by these findings, we conducted **additional qualitative analyses to explore the question: “What user-level factors most strongly influence personalization performance gains?”** By examining which users showed larger performance improvements between the two settings, we found that users with a **narrow information-seeking scope** exhibited significantly larger gains. These users repeatedly search or ask questions within a small set of topics, allowing their histories to represent clear and consistent preference signals. This makes it easier for the model to construct personalized responses.
>
> In contrast, users with **broad information-seeking scope**, spanning many diverse topics, showed small performance gains even with large numbers of sessions. Their preference signals are dispersed across multiple domains, and they often display different preference patterns depending on the topic. As a result, the model faces greater difficulty in determining which preference signals to apply for a given query, limiting the effectiveness of personalization.
>
> ## About ethics concerns
>
> This study was conducted under IRB approval, and the protection of participants’ rights and fair compensation were top priorities throughout the data collection pipeline. All annotators received detailed information about the procedures, compensation structure, and data usage before agreeing to participate.
>
> We provided each annotator with **$50 in monetary compensation**, and in addition, offered **three months of complimentary access to Google AI Pro Plan** after the data collection period for those who wished to continue using the account. To ensure participant autonomy, annotators were allowed to discontinue their annotation work at any point, whether during or after data collection. In such cases, they received **50% of the total compensation**, and **all data contributed by the participant were immediately and permanently deleted**.
>  This procedure ensured that annotators were never pressured to continue participation, while still receiving fair compensation for their contributions.
>
> All compensation schemes and procedures were reviewed in advance by legal experts to ensure compliance with ethical and legal standards.

---

### Author Response · Authors · 2025-12-02
**Rebuttal Summary Comment**

We are sincerely grateful to all reviewers for their positive assessment of our work and for providing constructive feedback. Below, we summarize how we addressed the concerns raised in the reviewers’ feedback.

### **1\) Ethical concerns about our dataset**

Reviewers expressed (AM4H, efh6) concerns about re-identification risk and compensation transparency. **We explained our IRB-approved multi-stage pipeline** including annotator self-scrubbing, multi-person manual review, repeated LLM screening, and conservative session-level removal. **We also clarified the specific compensation and voluntary withdrawal policy.** These are discussed in detail below:

- Initial response (3/3) for AM4H
- Initial response (2/2) for efh6
- Additional response for ethics concerns for efh6

### **2\) Our dataset is large in scale and sufficiently diverse**

Reviewers (AM4H, efh6) raised concerns about the dataset size and the adequacy of diversity. In our response, we clarified that **annotators were selected through a two-dimensional screening process covering diverse professional domains and interest areas**. We also explained that **our user interaction history contains a sufficiently large volume of interaction data**. These are discussed in detail below:

- Initial response (1/3) for AM4H
- Initial response (1/2) for efh6

### **3\) Open-weight models can also serve as effective evaluators**

Reviewers (AM4H, efh6) raised concerns centered on the heavy reliance on GPT-5 for rubric generation and evaluation, including potential model-family bias and reproducibility issues. We addressed this by evaluating two open-weight models as evaluators, demonstrating that **our benchmark generalizes beyond proprietary models and that evaluation cost can be substantially reduced.** These are discussed in detail below:

- Initial response (1/3) for AM4H
- Initial response (1/2) for efh6

### **4\) Motivation and objectives of our work in detail**

Reviewer (RMtA) requested further clarification regarding the motivation and goals of our work to achieve a deeper and clearer understanding of our contribution. **In response, we elaborated on these points based on the contents described in our paper, and additionally clarified how each component of our dataset and benchmark plays a role in supporting these goals.** These are discussed in detail below:

- Initial response (1/2) for RMtA
- Initial response (2/2) for RMtA

### **5\) Additional details and further in-depth analyses**

We incorporated the reviewers’ feedback by **providing additional implementation details and further in-depth analyses, and we reflected these updates in the revision.** We additionally reflected other detailed feedback provided by the reviewers. Below is a list of the additional details, analyses, and feedback we have incorporated:

- Data collection instructions and detail about quality control (AM4H, RMtA)
- Influence of backbone model capabilities (efh6)
- Impact of search quality on reasoning vs. non-reasoning models (eGSd)
- Feedbacks for improving the flow and clarity (AM4H, eGSd)


We hope that this summary helps consolidate how each concern has been addressed, and we believe that our revisions and additional analyses have further strengthened and deepened the contribution of our work.

---

### Meta-Review · Area_Chair_nvsR · 2026-01-11

**Summary:**

The paper proposes BESPOKE, a benchmark intended to evaluate personalization for search-augmented LLMs in information-seeking tasks. It collects (i) authentic user chat/search histories from 30 users over roughly three weeks (reported as 2870 history sessions), and (ii) 150 user-authored queries paired with gold information needs plus human scalar ratings and diagnostic feedback over multiple candidate model responses. The paper also introduces an LLM-based evaluation pipeline that generates query-specific rubrics and uses an LLM judge to score responses along dimensions such as need alignment, content depth, tone, and style, and reports analyses/ablations over context construction (e.g., general vs query-aware profiles, history selection) and the interaction between retrieval quality and model reasoning.

### Strengths
- The work targets an important evaluation gap: personalization for search-augmented LLMs in information-seeking scenarios.
- Data collection includes authentic histories and human-written diagnostic feedback, which is valuable for analysis beyond single-number ratings.

### Weaknesses
- Limited benchmark scale at the level of evaluation queries (150 total), raising concerns about coverage and statistical reliability of conclusions about personalization.
- Substantial reliance on proprietary LLMs throughout the pipeline (rubric generation, evaluation, and other pipeline components as described by reviewers), which undermines reproducibility and makes it difficult for the community to rerun/extend results end-to-end.
- The evaluation/pipeline description was viewed by reviewers as underspecified in the submission (particularly the operationalization in Section 4.2 and details of profile generation and data collection instructions/QC), making it hard to independently verify the setup.

**Reviewer Concerns:**

Addressed:
- Dataset diversity documentation: authors provided additional details about annotator screening by expertise/interest and summary statistics (e.g., domain/interest breadth, equitability).
- Pipeline clarifications: authors provided additional descriptions of annotator instructions and QC, specified k=6 for a sampling hyperparameter, and gave a cost breakdown.

Still outstanding:
- Scale/coverage: the rebuttal does not change the core limitation that the benchmark has only 150 evaluation queries, which remains the central concern for a benchmark meant for broad evaluation.
- Reproducibility end-to-end: while open-weight evaluators were tested, the overall pipeline remains described as relying on proprietary models in multiple stages; the rebuttal mitigates but does not eliminate the reproducibility issue.
- Ethics/privacy: authors described multi-stage de-identification and IRB oversight and clarified compensation, but the re-identification risk inherent to releasing interaction histories remains a substantive concern and not merely a documentation gap.

**Reviewer Scores:**

Likely unchanged.

---

### Decision · Program_Chairs · 2026-01-26

Reject